# Dynamic interplay of autophagy and membrane repair during *Mycobacterium tuberculosis* Infection

**Jacques Augenstreich** ⬤*, **Anna T. Phan, Charles N. S. Allen, Anushka Poddar, Hanzhang Chen, Lalitha Srinivasan, Volker Briken** ⬤*

Department of Cell Biology and Molecular Genetics, University of Maryland, College Park, Maryland United States of America

* jaugenst@umd.edu (JA); vbriken@umd.edu (VB)

## Abstract

Autophagy plays a crucial role in the host response to Mycobacterium tuberculosis (Mtb) infection, yet the dynamics and regulation of autophagy induction on Mtb-containing vacuoles (MCVs) remain only partially understood. We employed time-lapse confocal microscopy to investigate the recruitment of LC3B (LC3), a key autophagy marker, to MCVs at the single cell level with our newly developed workflow for single cell and single MCV tracking and fluorescence quantification. We show that approximately 70% of MCVs exhibited LC3 recruitment but that was lost in about 40% of those MCVs. The LC3 recruitment to MCVs displayed a high variability in timing that was independent of the size of the MCV or the bacterial burden. Most notably, the LC3-positive MCVs did not acidify, indicating that LC3 recruitment does not necessarily lead to the formation of mature autophagolysosomes. Interferon-gamma pre-treatment did not affect LC3 recruitment frequency or autophagosome acidification but increased the susceptibility of the macrophage to Mtb-induced cell death. LC3 recruitment and lysotracker staining were mutually exclusive events, alternating on some MCVs multiple times thus demonstrating a reversible aspect of the autophagy response. The LC3 recruitment was associated with galectin-3 and oxysterol-binding protein 1 staining, indicating a correlation with membrane damage and repair mechanisms. ATG7 knock-down did not impact membrane repair, suggesting that autophagy is not directly involved in this process but is coregulated by the membrane damage of MCVs. In summary, our findings provide novel insights into the dynamic and variable nature of LC3 recruitment to the MCVs over time during Mtb infection. Our data does not support a role for autophagy in either cell-autonomous defense against Mtb or membrane repair of the MCV in human macrophages. In addition, the combined dynamics of LC3 recruitment and Lysoview staining emerged as promising markers for investigating the damage and repair processes of phagosomal membranes.

**Data Availability Statement:** The source code for the pyimagej workflow for single MCV quantification is provided as supplementary material. The movies referenced in this article were

deposited here: https://data.mendeley.com/datasets/cvk8wnfm36/.

**Funding:** National Institute of Allergy and Infectious Diseases (Grant R01AI139492 to VB). The funders had no role in study design, data collection and analysis, decision to publish, or preparation of the manuscript.

**Competing interests:** The authors have declared that no competing interests exist.

## Author summary

Autophagy is a cellular process that allows cells to digest internal materials and may help in the defense against pathogenic bacteria, including *Mycobacterium tuberculosis* (Mtb), the causative agent of tuberculosis in humans. However, the precise dynamics of autophagy during Mtb infection remain unclear. This study investigated the interactions between LC3, a key component of the autophagy machinery, and mycobacteria-containing vacuoles (MCVs) within host cells. Using live-cell microscopy, we observed that approximately 70% of MCVs recruited LC3, with 40% of these subsequently losing LC3 association. The timing of LC3 recruitment varied considerably and showed no correlation with MCV size or bacterial load. Contrary to expectations, LC3 recruitment did not lead to increased acidification of MCVs, a process typically necessary for bacterial killing. Instead, we observed alternating patterns of LC3 and acidification markers on MCVs, suggesting a dynamic interplay between these processes. LC3 often appeared along with other signs of damage to the MCV membrane, suggesting it might be related to repair processes, but we did not find a role for autophagy in damage repair. In conclusion, our research gives us new insights into how autophagy interacts with Mtb over time. It suggests that this system might not directly fight Mtb or repair damaged MCVs in human cells.

## Introduction

*Mycobacterium tuberculosis* (Mtb) is the causative agent of the pulmonary disease tuberculosis, which is one of the deadliest bacterial infections worldwide (WHO) [1]. One of the main niche of replication for Mtb during infection is the macrophage [2–5]. An important characteristic of intracellular Mtb is its ability to evade host cell microbicidal responses [6–8]. One mechanism of the cell-intrinsic host defense is the targeting of the phagocytosed bacteria by the host cell autophagy to create an autophagosome through a process called xenophagy [9]. An important marker for autophagy activation is the protein microtubule-associated proteins 1A/1B light chain 3B (LC3), which is recruited to the membrane of the forming phagophore for the final formation of the autophagosome [9]. The recruitment of LC3 to the Mtb phagosomes can also happen independently of the canonical autophagic machinery in a process called LC3-associated phagocytosis (LAP) [10]. There is evidence that xenophagy limits Mtb proliferation in murine and human macrophages [11–13]. Nevertheless, the role of xenophagy in cell autonomous defense against Mtb during *in vivo* infections is more complex with some reports showing a more indirect role of autophagy for *in vivo* host resistance [14,15]. LAP also has a protective role in murine macrophages [16], but its capacity to inhibit Mtb replication in human macrophages has not been clearly demonstrated.

Even though they both require autophagy machinery, xenophagy and LAP are triggered in distinct ways. During Mtb infections, xenophagy is mainly induced by membrane damage to the Mtb-containing vacuole (MCV), which allows the recruitment of galectin proteins that recruit proteins of the autophagic machinery [12,17–20]. In contrast, LAP is independent of membrane damage induction and relies on TLR signaling [21], and LC3 conjugation to the phagosome is dependent on the production of reactive oxygen species (ROS) by the NOX2 complex [22]. Ultimately, both LAP and xenophagy activation is followed by phagosome/autophagosome maturation [10]. The maturation is mainly reflected by acidification of the autophagosome which is required for creating a bactericidal microenvironment and the activity of the lysosomal proteases and hydrolases [23]. This maturation occurs even though LC3 and autophagic machinery proteins were found to be dispensable for phagosome maturation

occurring independently of xenophagy as characterized by acidification or LAMP1 recruitment [24].

Mtb infection leads to the activation of autophagy signaling, but bacteria can inhibit steps of xenophagy and LAP to increase their survival within macrophages [7,8]. In the model system of *Dictyostelium discoideum* infected by *Mycobacterium marinum* (Mma), the phagosome escape by Mma triggers xenophagy, but the maturation of the autophagosome is blocked by the bacteria [25,26]. This inhibition of the maturation of the autophagosome was also observed in Mtb-infected human dendritic cells [27] and macrophages [28] and is, at least partially, mediated by the Mtb cell envelope lipid phthiocerol dimycocerosate (DIM/PDIM) [29]. The Eis and PE-PGRS47 proteins are among several Mtb proteins that have been identified to mediate the inhibition of host cell xenophagy (for review [7]) [30–33]. The observation of xenophagy via electron microscopy highlighted that Mtb could escape from the newly formed autophagosome [34]. Mtb inhibited LAP via CpsA; however, the inhibition of LC3 recruitment was rescued by priming the macrophages with interferon-Gamma (IFN-γ) [16].

Mtb manipulates many host cell signaling pathways in order to survive and replicate in macrophages [7,8]. A hallmark discovery was the observation that Mtb inhibits the maturation process of the MCV [35]. Many additional reports established a dogma that Mtb remains within an immature phagosome during its intracellular infection phase [36–40]. This dogma was overturned when it was shown that a fraction of the bacteria can escape the MCV into the cytosol [41,42]. Additional studies show that a fraction of MCVs do not inhibit phagosome maturation but acidify and obtain characteristics of phagolysosomes [41,43]. It is not uncommon for intracellular bacterial pathogens to occupy several intracellular niches [44]. What is the interaction of these different populations of bacteria within the infected cell and how does the niche that Mtb occupies determine its interaction with the host cell autophagy machinery? For example, the contact of the bacteria with the cytosol triggers ubiquitination of Mtb involving the E3 ligase Parkin and the DNA sensor STING, that can trigger xenophagy in an attempt to recapture the bacteria [12,19]. Autophagy induction is proposed to directly control Mtb infection in macrophages by increasing the killing of the bacteria in an mature autophagolysosome. However, with Mtb able to escape the autophagosome, and the mounting evidence that *in vivo*, the role of autophagy might be indirect via some immune system modulation, the question remains of how directly bactericidal autophagy is. To start answering these questions we performed a spatiotemporal analysis of the interaction of individual bacteria with the host cell LC3 protein as a marker for autophagy responses.

The crosstalk between Mtb and autophagy (*i.e* xenophagy, LAP) is highly dynamic and diverse, showing a duality between induction and inhibition of these responses by the bacteria. Live cell imaging coupled with time-lapse acquisition at the single-cell level is an excellent approach to investigate host pathogen interactions in time and space [45] but has only recently been applied to study Mtb infections [4,34,46–48,13]. In this study, we monitored the dynamics of LC3 association to the phagosome during the infection of THP-1 cells with Mtb using time lapse confocal microscopy. We also implemented our recently released workflow combining different methods of deep learning-based segmentation and rational tracking of cells to analyze fluorescence signals at the single-cell level and/or at the single bacteria level [49]. The results show that LC3 recruitment to the MCV is frequent but in many cases only temporary. We demonstrate that bacterial burden does not influence LC3 recruitment but may accelerate its occurrence in the cells showing the recruitment. Next, we observed that the priming of cells with IFNγ did not affect LC3 recruitment frequency nor acidification of the MCV. However, in most cases LC3 recruitment was preceded by a loss in acidification of the MCV followed by a restoration of acidity when LC3 is recruited suggesting a role of autophagy in membrane repair of the MCV. Nevertheless, knock-down approaches showed that LC3 recruitment is

actually not involved in membrane repair but is concomitantly activated with the endoplasmatic reticulum (ER) dependent membrane repair machinery.

## Results

### The recruitment of LC3 is frequently observed around the MCVs over the course of infection

Multiple studies have shown that Mtb can induce autophagy at specific time points post-infection. Autophagy was evaluated by immunofluorescence staining on fixed samples [18,29] or stable expressions of recombinant fluorescence proteins [34,46], but for the latter lacked clear quantification and timing determination. Consequently, these studies offered an incomplete view of the dynamic events and, for example, did not allow to determine if LC3 recruitment to the MCV follows a specific timing after phagocytosis or if it is a transient phenomenon.

To evaluate the dynamics of autophagy induction during infection with Mtb, we used PMA-differentiated THP-1 cells stably expressing GFP-LC3 [50] and infected them with a fluorescent strain of Mtb (H37Rv-DsRed). The cells were followed by time-lapse confocal microscopy during 16–20 hours after the addition of bacteria at on MOI of 2 (Fig 1A, Movie 1 at https://data.mendeley.com/datasets/cvk8wnfm36/). The LC3 recruitment to the MCV was visually assessed from the time of phagocytosis to the end of recording for individual MCV. The results showed that ~70% of MCVs triggered LC3 recruitment (Fig 1B) at one point or another after phagocytosis. However, almost half of the LC3-positive (LC3$^+$) MCVs became negative by the end of the recording showing efficient escape of Mtb from the LC3 recruitment. We also observed two patterns of escape from LC3 recruitment by Mtb (Fig 1C and 1D). One pattern (Exclusion) which is characterized by an initial increase of the LC3 signal at the MCV that quickly diffused away from the bacteria (Fig 1C, Movie 2 at https://data.mendeley.com/datasets/cvk8wnfm36/). The other pattern observed (Egress) is characterized by the accumulation of LC3 signal at the MCV with the formation of tubulovesicular structures (TVS) [51], then the bacteria and the TVS separate (Fig 1D, Movie 3 at https://data.mendeley.com/datasets/cvk8wnfm36/). An example of continuous recruitment of LC3 to the MCV is shown with the formation of an Mtb-containing autophagosome (Fig 1E, Movie 4 at https://data.mendeley.com/datasets/cvk8wnfm36/). Overall, LC3 recruitment seemed frequent around MCVs (~70% of all observed MCVs), but in a significant fraction of these recruitments (about 40%) the bacteria appear to escape from the response.

### No correlation between autophagy induction and the time after infection nor the size of the MCV

To elucidate the dynamic LC3-MCV interactions (Fig 1), we conducted a more detailed analysis of LC3 recruitment to MCVs. We analyzed a total of 73 MCVs showing at least one temporary LC3 recruitment by their time post-phagocytosis and their observed length of time of LC3 recruitment (Fig 2A). Three groups of MCVs could eventually be distinguished: (i) MCVs showing LC3 recruitment between 0 and 1 h post phagocytosis (hpp) (Fig 2A–2C), (ii) MCVs showing recruitment after 1 hpp and (iii) the MCVs presenting multiple LC3 recruitment events. For each group, an example is given (Fig 2B–2G), where the fluorescence quantification on the bacterial area over time (Fig 2C, 2E and 2F, Movies 4–6 at https://data.mendeley.com/datasets/cvk8wnfm36/) could confirm the visual observation as shown on representative snapshots from the time-lapse study (Fig 2B, 2D and 2F). We observed a 40-minutes gap between early and late recruitment events, meaning that in the early event category LC3 recruitment was always observed between 0 to 0.33 hpp and the recruitment in the late event category

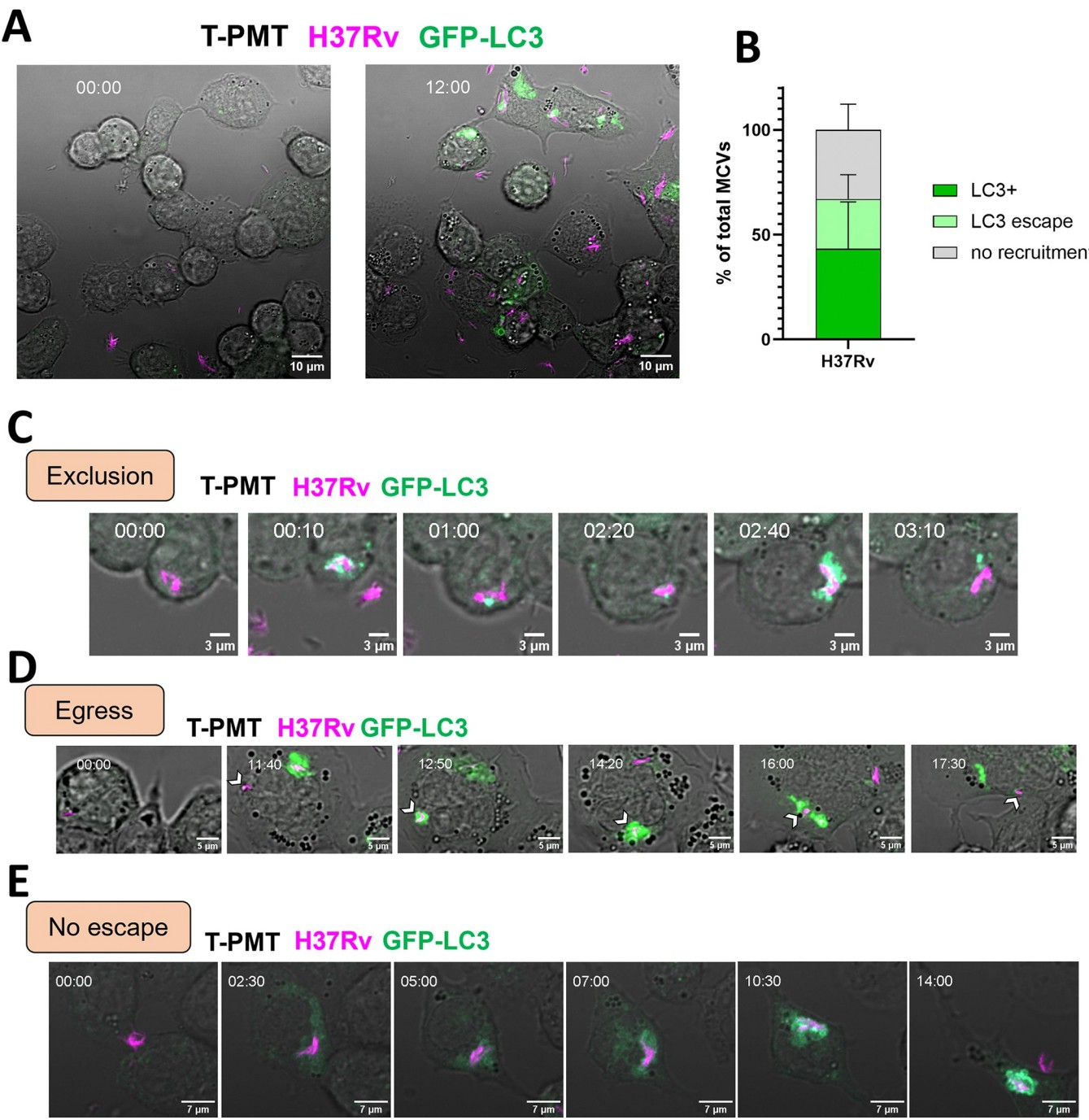

**Fig 1. The Mtb phagosome induces frequent LC3 recruitment but can escape from it.** THP-1 cells stably expressing GFP-LC3 were infected with DsRed-expressing Mtb H37Rv at an MOI of 2 for 16–20 h. A 9μm z-stack with a 1μm step was recorded every 10 minutes. (A) A representative image of infected THP-1 cells at the beginning of recording and at 12h post-infection. Time stamp format is hh:mm. (B) quantification of LC3 recruitment to mycobacteria-containing vacuoles (MCVs) (LC3+), and subsequent LC3 escape or no LC3 recruitment. The graph shows the mean and standard deviation from 4 independent experiments, following between 22–43 MCVs per experiment. (C) Representative time lapse images extracted from Movie 2 (at https://data.mendeley.com/datasets/cvk8wnfm36/) of LC3 recruitment to an MCV followed by loss of LC3 signal by exclusion. (D) Representative time lapse images extracted from Movie 3 (at https://data.mendeley.com/datasets/cvk8wnfm36/) of LC3 recruitment to an MCV followed signal by egress of the bacteria out of the autophagosome. (E) Representative time lapse of an MCV exhibiting a LC3 recruitment.

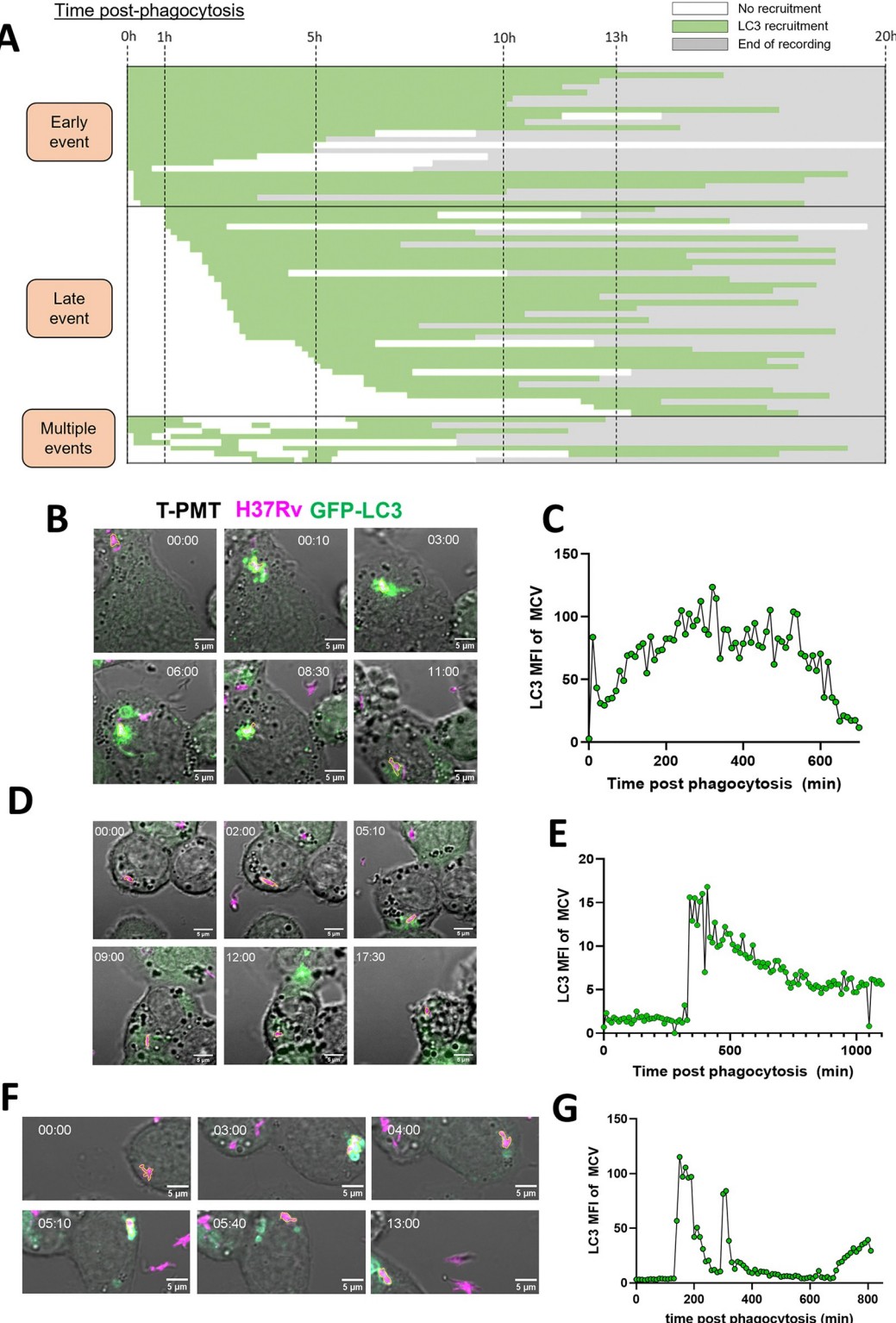

**Fig 2. LC3 recruitment does not follow a specific timing pattern.** (A) MCVs were categorized by their time of observed LC3 recruitment after phagocytosis. The green color indicates visually observed LC3 recruitment, and the white color no recruitment. The grey color indicates end of recording. The total number of MCVs observed was n = 73. MCVs were organized into 3 categories: 1. "Early events" the ones showing LC3 recruitment immediately or shortly after phagocytosis (≤1 h post phagocytosis), 2. "Late events" the MCVs showing an LC3 recruitment > 1 h post phagocytosis and 3. "multiple

events" the MCVs showing multiple LC3 recruitments. (B, D, F) Representative time lapses of MCVs of the 3 listed categories, extracted from Movies 4, 5 and 6 (at https://data.mendeley.com/datasets/cvk8wnfm36/) respectively. MCVs followed were outlines with a yellow region of interest (ROI) line. (C, E, G) Confirmation of visual recruitment by quantification of LC3 fluorescence intensity over time on the three MCVs in (B, D, F).

started after 1 hpp (Fig 2A). However, within the late events group no clear pattern emerged that would indicate that the LC3 recruitment is governed by a clear timing factor. Instead, we observed a wide range of LC3 recruitment times, from 1 to about 13 hpp. indicating an absence of a coordinate response involving LC3 after Mtb phagocytosis.

In summary, LC3 recruitment to MCVs is highly variable in timing, from early recruitments ($< 0.33$ hpp) to late recruitments ($> 1$ hpp), without clear patterns of a coordinated response.

A previous study proposed that the number of bacteria per MCV and more specifically their capacity to form cords could inhibit xenophagy in lymphatic endothelial cells [52]. Intracellular cord formation is not occurring in differentiated THP-1 cells and thus could not be a factor in our system but the influence of bacterial burden per cell on LC3-recruitment was assessed. First, we compared the number of bacteria per MCVs to the time post infection when the first LC3 recruitment was observed (Fig 3A). A modest negative correlation can be observed between the number of bacteria per MCV and the time of recruitment of LC3 to the MCV (R = -0.2671, $p$ = 0.0315). However, this result was not consistent with our result that on average, the number of bacteria per MCV was not different between the early and late recruitment groups or the group with multiple recruitment events (Figs 2A and 3B).

## The bacterial burden does not directly drive LC3 recruitment but may accelerate its occurrence

Even though the size of the MCV itself did not appear to strongly correlate with initiation of LC3 recruitment, in some infected cells, it seemed that subsequent LC3 recruitment happened because other MCVs were already present inside the cell. For example, in this recording of the cells (Fig 1D, Movie 3 at https://data.mendeley.com/datasets/cvk8wnfm36/), the first bacteria entered the cell, and no LC3 recruitment to the MCV was observed for 11 hours and 50 minutes. Then a second bacterium was phagocytosed and immediately ($\leq 10$ min) triggered LC3 recruitment to that MCV. Interestingly, in this cell even the first MCV underwent LC3 recruitment shortly after the second MCV became LC3$^+$. This series of events suggested that the second MCV triggered a response that then targeted both MCVs.

To analyze this phenomenon on a larger number of single cells, we implemented the new analysis method that we recently released [49] on the different time-lapsed acquired. With this method, the cells that would stay in the field of view were tracked, isolated and separated into 3 groups: Uninfected, infected with LC3 recruitment on at least one MCV, and infected without visible LC3 recruitment on any MCV. The bacterial burden was calculated following the BBQ method recently released based on the relative quantification by bacterial volume calculation. To compare the level of bacterial burden over time with the level of autophagy induction, an analysis based on the volume of the LC3$^+$ tubulovesicular structure was done based on the bacterial volume calculation logic (see methods). To validate that this approach is a relevant readout, the LC3 volume over time from single cells was pooled and plotted for the different cell populations (S1 Fig). This confirmed that in the infected cells that visually showed LC3 recruitment, the LC3 volume consistently increased over time (S1A Fig), whereas in the infected cells without visual autophagy induction, no or barely and LC3 volume increase was measured (S1B Fig). Some increased LC3 volume increase was also occasionally observed in

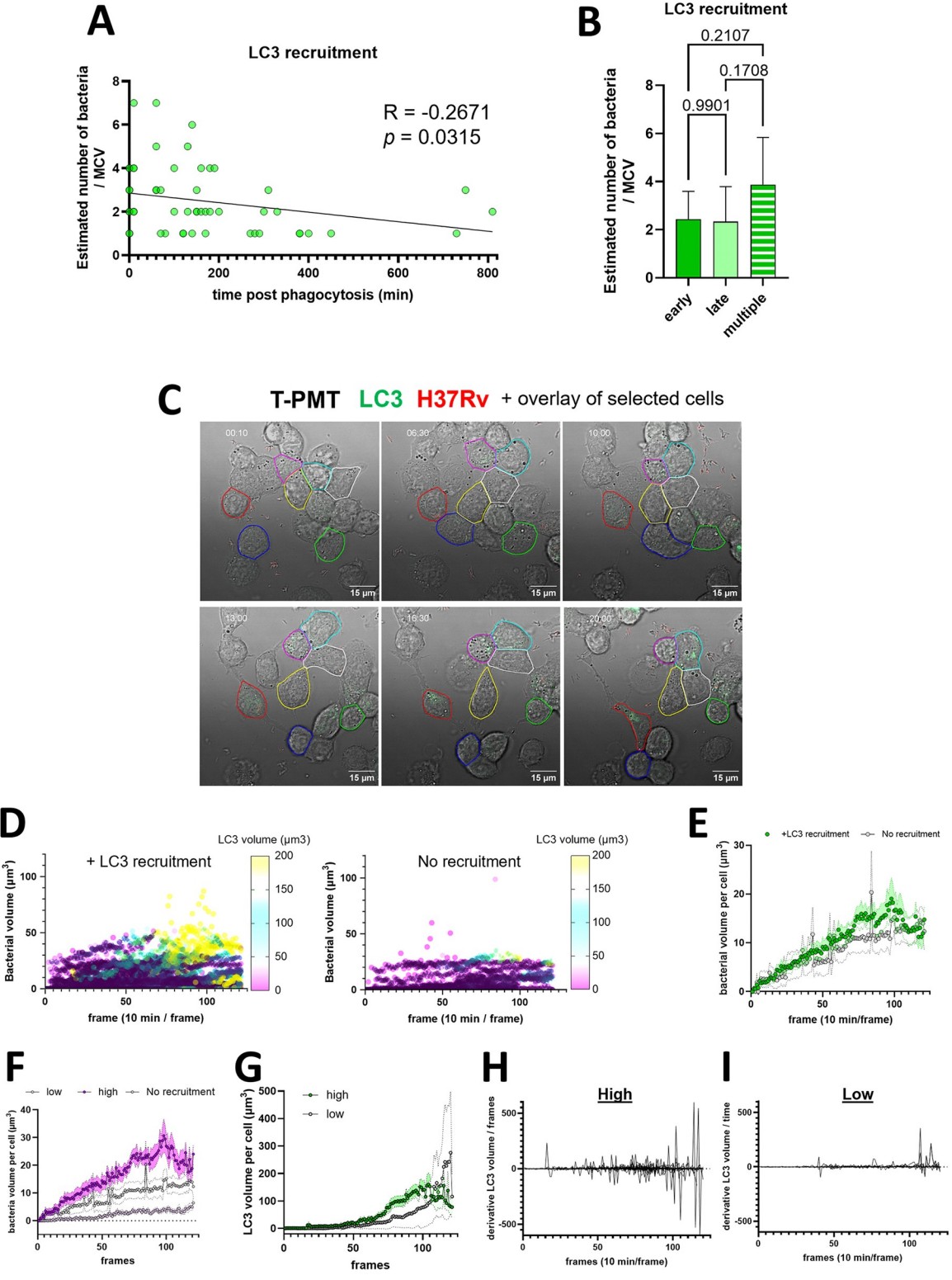

Fig 3. Macrophages bacterial burden does not directly influence the LC3 recruitment to MCVs. (A) Distribution of number of bacteria per MCVs compared to the time of LC3 recruitment. The correlation between the 2 variables was tested using the spearman correlation test. The correlation coefficient R and the *p*-value is displayed on the graph. (B) Quantification of the average number of bacteria per MCV in each group from Fig 2A. (n = 29 early event, n = 36 late event and n = 8 multiple events). The groups were compared by one-way ANOVA test followed by Dunnett *post-hoc* statistical test. (C) Representative time lapse with the overlay showing

an example result of the single cell tracking, selection of cell of interest outlined and quantification described in methodology. (D) Single cell tracking and quantification of bacterial volume over time plotted on a 3 variables graph with the LC3 volume as a color dimension. Each dot represents the measurement in 1 cell in 1 frame. The left panel represents the quantification in cells exhibiting at least one temporary LC3 recruitment on at least one MCV ($N_{cells}$ = 41, $N_{points}$ = 4299). The right panel represents the quantification in cells that don't exhibit LC3 recruitment ($N_{cells}$ = 11, $N_{points}$ = 1222). (E) Quantification of the average bacterial burden in $\mu m^3$ over time, on the cell population described in (D). The dots represent the mean and the band around the connecting line the standard error to the mean. (F) Quantification of the average bacterial burden in $\mu m^3$ over time, on the cell population exhibiting a high (>20 $\mu m^3$, $N_{cells}$ = 24) or low (<20 $\mu m^3$, $N_{cells}$ = 17) bacterial burden among the cells plotted in (D-left). (G) Calculation of the associated LC3 volume in the high and low cells. (H-I) The first derivative of the LC3 volume curves of each individual cells of the 'high' (H) or 'low' (I) groups was calculated and plotted. All the data here from 4 independent experiments.

some uninfected cells (S1C Fig) but was associated with an increase in the overall signal of LC3 (Movie 1 at https://data.mendeley.com/datasets/cvk8wnfm36/, uninfected cell in the center). From this, the bacterial volume was plotted over time with the measure LC3 volume as a 3rd color dimension (Fig 3D). The increase in bacterial burden can be seen but seemed at least partially higher for the infected exhibiting LC3 recruitment. The displayed color for the LC3 volume seemed also in the brighter colors for the cells exhibiting a higher bacterial burden, meaning a higher LC3 signal in these cells. To deepen the analysis on the potential role of the bacterial burden in the induction of autophagy, the average bacterial burden over time between the 2 cells groups was calculated (Fig 3E). On average, there was no difference of burden between the cells with LC3 recruitment compared to the cells without. The absence of evidence from the influence of the bacteria may suggest that the deciding factor of autophagy may come from the host side, where a cell to cell variability may occur, like in the case of iNOS [47]. We then decided to focus on the cells showing an induction of autophagy and hypothesized that in this population the burden may influence the autophagy dynamics. The cells were separated in two groups, the one reaching a bacterial volume ≥20$\mu m^3$ (high) or <20 $\mu m^3$ (low) (Fig 3F) and their respective average LC3 volume over time was plotted (Fig 3G). The results showed a noticeable trend on the LC3 recruitment being induced earlier in the high burden population compared to the low burden, before reaching an equivalent level. To visualize this phenomenon in a single cell fashion, the individual curves were transformed into their partial derivative calculated between each time points, plotted and superimposed by their respective cell category (Fig 3H and 3I). The derivative over time indicates the changes in the slop of the curve, thus, should be able to display the sudden changes of LC3 volume that could happen in case of autophagy induction. Accordingly, in the cell population of high burden, the derivative showed a higher amplitude over time (Fig 3H), with a trend of LC3 volume changes earlier than for the cell population with low burden (Fig 3I). These results then strongly suggest that even if the burden is not a direct driver of LC3 recruitment, a higher burden may accelerate its occurrence.

## LC3+ MCVs do not acidify, and the recruitment frequency is not affected by IFN-γ pre-treatment of macrophages

During the time-lapses, a non-negligible number of bacteria did not show signs of escaping LC3 association (Fig 1B) suggesting that they might be contained in autophagosomes that could mature and eventually kill the bacteria. To explore the fate of LC3+ MCVs, the cells were infected for 4h and later dyed with Lysotracker blue (LTB) to assess phagosome or autophagosome acidity (Fig 4A). Strikingly, at 24 hours post-infection, none of the LC3+ MCVs were positive for LTB, although about 27% of MCVs were positive for LTB (Fig 4B). These results strongly suggested that LC3 recruitment mainly does not result in Mtb contained within autophagolysosomes. Previous studies showed that pretreatment with IFN-γ increased LC3 association with MCVs and phagosome maturation [5,16]. To study this in a live-imaging

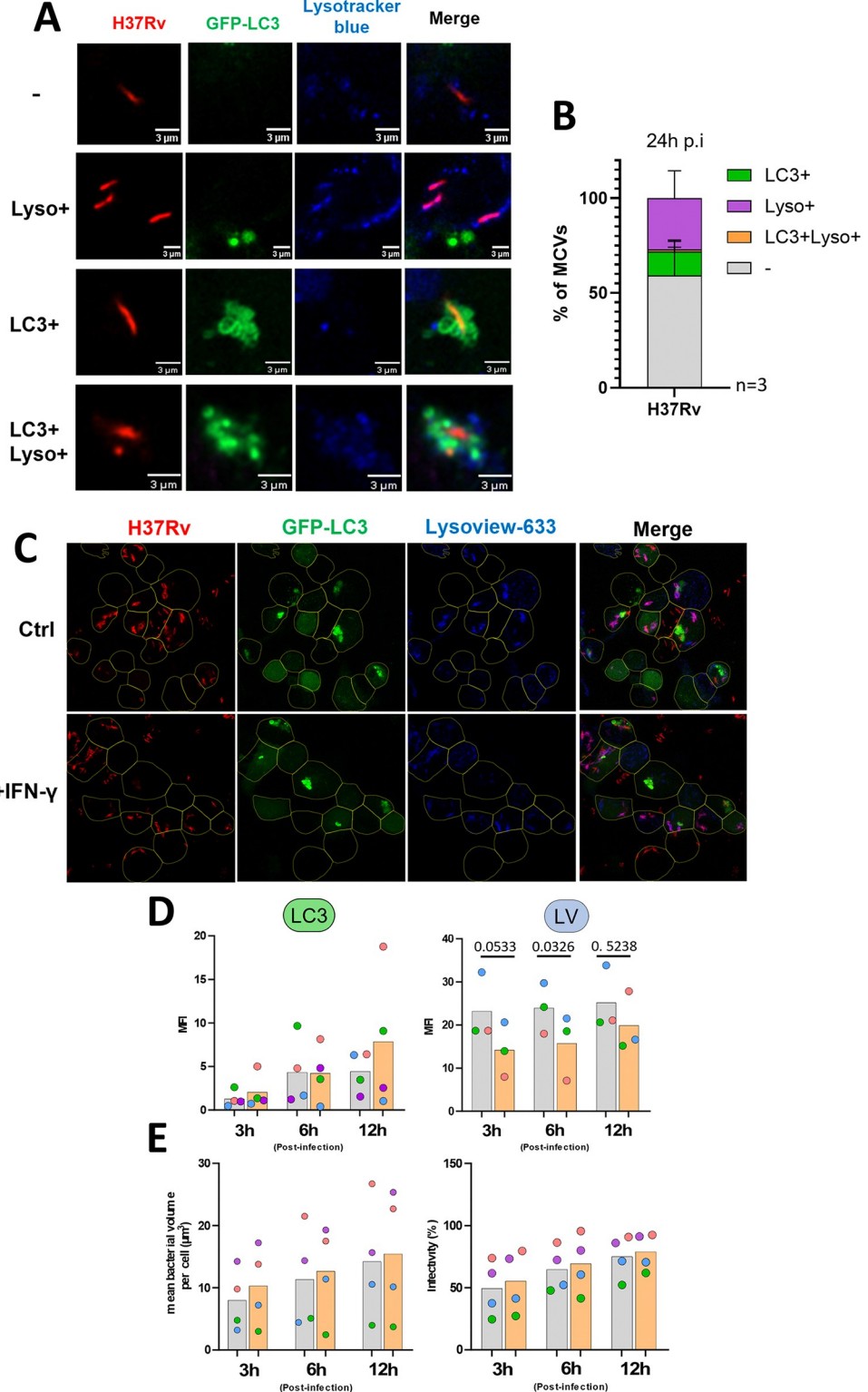

**Fig 4. Autophagosomes do not acidify and IFN-γ treatment has no effect on LC3 recruitment frequency or autophagosomes acidification.** (A-B) THP-1 expressing GFP-LC3 were infected at an MOI of 10 for 4 h and incubated for 24 h. Cells were then stained with Lysotracker blue and imaged by confocal microscopy. (A) representative image of MCVs that were unstained, positive for LC3 alone (LC3+), positive for lysotracker alone (Lyso +), or double positive (LC3+Lyso+). (B) quantification of LC3 and Lysotracker staining on MCVs. The data were

obtained from 3 independent experiments. (C-E) THP-1- GFP-LC3 cells were treated with IFN-γ 10ng/mL or left untreated overnight and infected with DsRed expressing H37Rv at MOI 2. Cells were also stained with Lysoview-633 (LV) and the dye was kept during image acquisition. Cells were imaged by time lapse confocal microscopy. (C) Representative images at 6 h post infection showing the untreated cells (top panel) or pre-stimulated with IFN-γ (bottom panel). The segmented, alive cells are outlined in yellow. (D) Quantification of the mean fluorescence intensity (MFI) of GFP-LC3 (left) and Lysoview (right) on single MCVs in living cells. Data points represent the average MFI on MCVs from 3 to 4 independent experiments. The color of the points indicates data obtained from the same experiment. (E) Quantification of bacterial burden, infectivity using the volume calculation as described elsewhere.[49] Data are from 4 independent experiments. Data points color indicate the same experiment. Data between untreated and IFN-γ treated groups were compared by paired *t*-test.

condition, we opted for the far-red probe Lysoview-633 (LV), which showed a similar pattern to LTB at 24 hours post-infection (hpi) and was not cytotoxic when left in the medium for 48 hours (S2 Fig). To quantify the signals on the MCVs, we implemented a new workflow for high throughput, single MCV fluorescence quantification (see methods, S3 Fig). The result of the pretreatment of THP-1-GFP-LC3 cells with 10 ng/mL of IFN-γ overnight showed that on average LC3 fluorescence on MCV was not different in cells treated or not with IFN-γ at all the analyzed time points, indicating that the treatment had no effect on LC3 recruitment (Fig 4D, left). And the LV was also found decreased, specifically at 1 h and 6 hpi in IFN-γ treated cells compared to untreated (Fig 4D, right). However, pretreatment with IFN-γ made the cells more susceptible to Mtb-induced cell death, characterized by "ballooning" morphology as observed through the transmitted light channel (S4B Fig, Movies 9–10 at https://data.mendeley.com/datasets/cvk8wnfm36/). The quantification of this phenomenon at 20 hpi showed an increase from 40% in untreated cells to 75% in IFN-γ pretreated cells after Mtb infection (S4C Fig). To explore the eventuality that this death could be due to difference in bacterial load or infectivity, these parameters were measured at 6 hpi. The results showed that no notable difference was observed between treated and untreated cells (Fig 4E). In summary, our results showed that LC3 recruitment almost never turned into a mature autophagolysosome, and that IFN-γ was not able to increase the frequency of LC3 recruitment or the maturation level of the autophagosomes.

## LC3 recruitment and acidification of the MCVs are mutually exclusive events

Our steady state analysis showed few MCVs (between 0% and 1.96% of the MCVs analyzed) that were positive for LC3 and lysotracker staining (Figs 4B and S2). From the untreated cells in Fig 4C–4E, we selected MCVs that showed LC3 recruitment and could be followed for at least 3 h to investigate the dynamics of LV and LC3 association with the MCV. The analysis showed that LC3 and LV staining does not happen simultaneously (Figs 5A–5D and S4). Instead, we observed acidification of the MCV, followed by a drop of acidification prior to the subsequent LC3 recruitment. In some cases, the LC3 signal dropped and was followed by a quick recovery in acidification thus completing a cycle that could be repeated (Fig 5A and 5B, and Movie 11 at https://data.mendeley.com/datasets/cvk8wnfm36/). In other cases, the LC3 recruitment would persist for prolonged periods of time (>6 h) until the end of the recording (Fig 5C, 5D, Movie 12 at https://data.mendeley.com/datasets/cvk8wnfm36/). We did not observe cases in which the MCVs exhibit LC3 recruitment without first acidifying. We also did not observe cases in which the Lysoview signal on the MCVs dropped without gaining LC3 staining. To further visualize the link between acidification and LC3 recruitment, we plotted the LC3 signal from single MCVs against the LV fluorescence at every time point recorded (Figs 5E and S5). The resulting dot plot and density map clearly show two populations of stained MCVs, the ones exhibiting signal for LC3 (from 40–100% of normalized intensity),

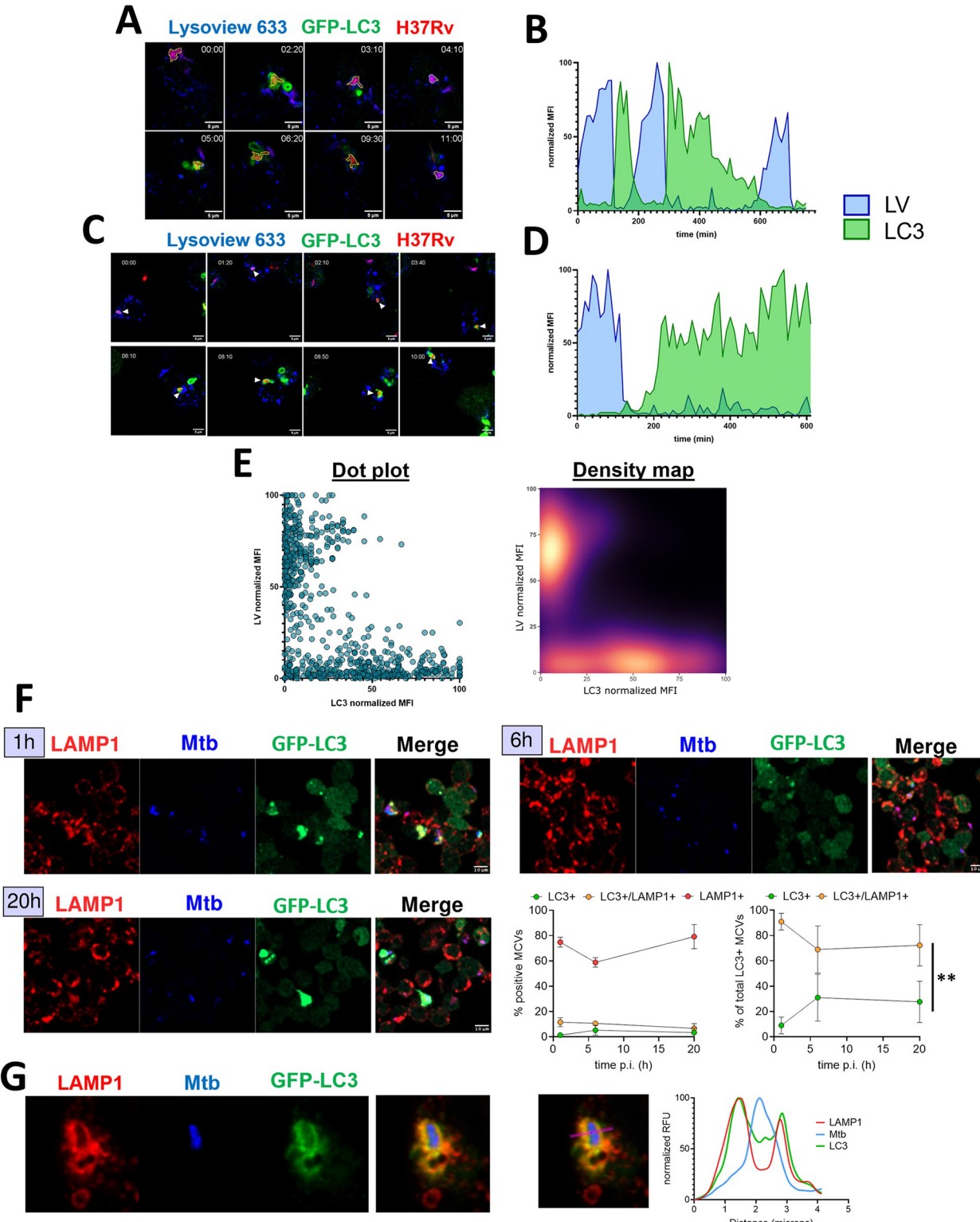

**Fig 5. LC3 recruitment and phagosome acidification are mutually exclusive, but LC3-TVS can fuse with late endosomes / lysosomes.** Single MCV tracking in unstimulated THP-1- GFP-LC3 cells infected with DsRed expressing H37Rv in Fig 4. (A, C) representative time lapses of MCVs outlined with a yellow ROI. (B, D) quantification of fluorescence at the MCV for each time point. Mean fluorescence intensity of GFP and Lysoview was measured and normalized between values of 0 and 100 as 0 the minimum value recorded, and 100 the maximum value. (E, left panel) Quantification of Lysoview and LC3 normalized MFI and was compared. Each dot represents the GFP and LV normalized MFI at one time point

for one MCV from 3 independent experiments (n = 12 MCVs). The individual fluorescence traces of each MCVs are shown in S4 Fig. (E, right panel) The distribution of data points in left panel is shown as a density map. (F-G) THP-1-GFP-LC3 cells were infected at an MOI 10 for 4 h, and incubated for 1 h, 6 h and 20 h post infection. At each time the cells were fixed and stained for LAMP1. The cells were imaged by standard confocal microscopy (F) or by Airyscan confocal microscopy (G). (F) Representative images of infected cells at the designated time point. The quantification of association of LC3 and/or LAMP1 over time is shown. The left graph shows the raw percentage of association of LAMP1 alone (LAMP1+), LC3 alone (LC3+) or both (LC3+/LAMP1+). The right graph shows the proportion of MCV with LC3 alone (LC3+) or LC3 with LAMP1(LC3 +/LAMP1+) among the total amount of MCV being LC3+. The graphs show the mean and standard error from two independent experiments. The curves were compared using the modified chi-square comparison. (G) Representative image obtained by Airyscan microscopy. (right) quantification of fluorescence intensity on the on area of interest in purple (bar 5 pixels of width). The fluorescence intensity was normalized by their maximum and minimum values and plotted.

and ones showing a signal for LV (from 40–100% of normalized intensity) with very rare overlap between the 2 markers (Fig 5E). As these results are observed in THP-1 cell line, we wondered if this interplay between LC3 recruitment and acidification could be observed in primary human macrophages. We generated human monocyte-derived macrophages (hMDMs) and transfected them to express GFP-LC3. The macrophages were infected and imaged identically to the THP-1 cells. Similarly, we observed that a drop of Lysoview signal would precede the recruitment of LC3 on the MCV (S7A Fig). Similarly, the analysis of the overall signal of every tracked MCV revealed populations of MCV having fluorescence of LC3 or LV but rarely both.

This result confirms the observation that LC3 structures are rarely acidic and that an acidified vacuole loses its acidification before LC3 recruitment, indicating that membrane damage occurred and was recognized by the autophagic machinery. The concurrent recovery of LV staining and loss of LC3 signal suggests potential involvement of autophagy machinery in membrane repair of the MCV.

The absence of acidification may indicate defective autophagosome maturation. To evaluate whether LC3$^+$ MCVs were fusing with late endosomes or lysosomes, a pulse-chase infection assay was performed, followed by immunostaining at various post-infection time points. Cells were then imaged by confocal microscopy (Fig 5F). To analyze LAMP1 and LC3 recruitment, we implemented a spatial and quantitative fluorescence analysis to classify the signals based on their distribution on and around the MCVs. The MCVs were automatically detected using Trackmate and Fiji similarly to the workflow described in S3 Fig (see methods, S6 Fig). The recruitment of the markers on a MCV was determined by measuring the fluorescence intensity on the MCV every 200nm in 1μm range. This creates a curve of fluorescence intensity for each detected MCV whose shape reflects the accumulation or not of markers on or around the MCV (S6 Fig). The pool of curves was then classified using an agglomerative clustering algorithm and grouped into "positives" or "negatives" for the analyzed marker. The presence of both LC3 and the second marker analyzed was validated if the MCV would be classified as positive for both markers. The results showed that at all post-infection time points, most LC3-positive MCVs were also positive for LAMP1 (Fig 5F). A higher-resolution analysis using Airyscan microscopy further revealed overlap between LC3 structures and LAMP1 staining, suggesting that the phagosomal membrane and LC3-TVS are undergoing phagosome-lysosome fusion (Fig 5G). Therefore, the lack of acidification observed in live time-lapse imaging likely reflects phagosomal maturation that fails to result in acidification as detectable by LV staining.

## Autophagy is associated with membrane damage and repair effectors but is not involved in membrane repair

As the dynamics of GFP-LC3 and LV strongly suggested the presence of damages on the phagosomal membrane, we tested the involvement of EsxA in the LC3 recruitment as documented

elsewhere [34]. As expected, the quantification of autophagy by measuring the volume of LC3 in tracked single infected cells showed that the induction of autophagy was observed in majority with wild-type (WT) bacteria compared to bacteria deficient in *esxA* (ΔesxA) (S7C-S7D Fig). In the rare case of LC3 recruitment to the ΔesxA MCV, we observed the same mutually exclusive staining of LC3 and LV on the MCV as for Mtb (S7E-S7F Fig). Mtb was also found to counteract phagosome maturation and acidification, which could explain for a loss of signal of Lysoview. MCVs of Mtb WT and ΔesxA that would not show LC3 recruitment were tracked to analyze the progression of LV staining over time. We observed that the initial increase of LV staining was slower for the WT strain compared to the ΔesxA. Ultimately, the LV signal stabilizes to a similar value between the two strains. These results support the conclusion that the loss of LV staining and subsequent LC3 recruitment is due to membrane damages of the MCV that are dependent on EsxA expression by Mtb.

Next, we evaluated the recruitment of markers of membrane damage and various membrane repair pathways to the MCV. For these experiments, the cells were infected for 4 h and incubated for the designated times. Confocal immunofluorescence microscopy was used to detect Galectin-3 (GAL3), a marker of membrane damage induced by Mtb [53], CHMP4B, a core component of the ESCRT-III machinery that was recently observed to be recruited to the MCV [54]. and finally the oxysterol-binding protein 1 (OSBP) involved in ER-dependent membrane repair recently described in *Mycobacterium marinum* phagosome repair and recruited to the Mtb phagosome [55]. The antibodies targeting the targets mentioned above were validated using LLOMe to induce lysosome damage and LC3 recruitment to the damaged compartment (S8 Fig). The cells were then infected and at 1 h, 6 h and 20 h post infection, the cells were stained for either of the markers and imaged by confocal microscopy. The recruitment of LC3 and the second marker was analyzed as described for LAMP1 (S6 Fig). The results first confirmed that LC3 recruitment is frequently associated with GAL3 recruitment as previously described by others (Fig 6A and 6B) [46]. However, the LC3$^+$ structures were more rarely associated with the effector CHMP4B compared to GAL3 over the different time points, also showing that ESCRT-III machinery was recruited to the MCV but mostly not together with LC3 recruitment in these experiments (Fig 6A and 6B). LC3$^+$ MCVs were in the majority co-stained with OSBP at all time-points. (Fig 6A and 6B), thus showing that LC3 recruitment is indeed associated with some membrane repair effectors and that the acidification recovery observed previously (Figs 5A, 5B and S4) corresponds to the phagosomal membrane getting repaired, most likely through an OSBP-dependent, ER-membrane repair machinery.

The next question was to determine if LC3 recruitment and OSBP are directly involved in the membrane repair that was previously observed. To answer it, ATG7 and OSBP were knocked-down using siRNA (S9 Fig), and the cells infected with Mtb and followed by time-lapse confocal imaging. In case of a direct involvement of autophagy in membrane repair we would expect to see a decrease in the average signal of lysoview on the bacteria as there would be an accumulation of damaged and unrepaired phagosomes which are LV$^-$. If the OSBP knock-down (KD) affect membrane repair, we would expect to see an increased LC3 recruitment as an accumulation of damaged compartment would increase the frequency of autophagy induction around bacteria. Thus, the mean fluorescence intensity of LC3 and lysoview was measured on single bacteria at different time post-infection (Fig 6C and 6E). For OSBP-KD, the silencing resulted in at least a temporary increase in LC3 recruitment on the MCVs, while the LV tended to decrease (Fig 6C). However, in the ATG7-KD condition, while the silencing reduced LC3 recruitment, the lysoview staining was at least temporarily increased on the MCVs (Fig 6E). To alleviate any effect due to bacterial burden (Fig 3D), the effect of the respective KD on bacterial burden and infectivity was measured (Figs 6D and 6F, and S10). The result showed no clear differences in intracellular bacterial growth nor the infectivity over

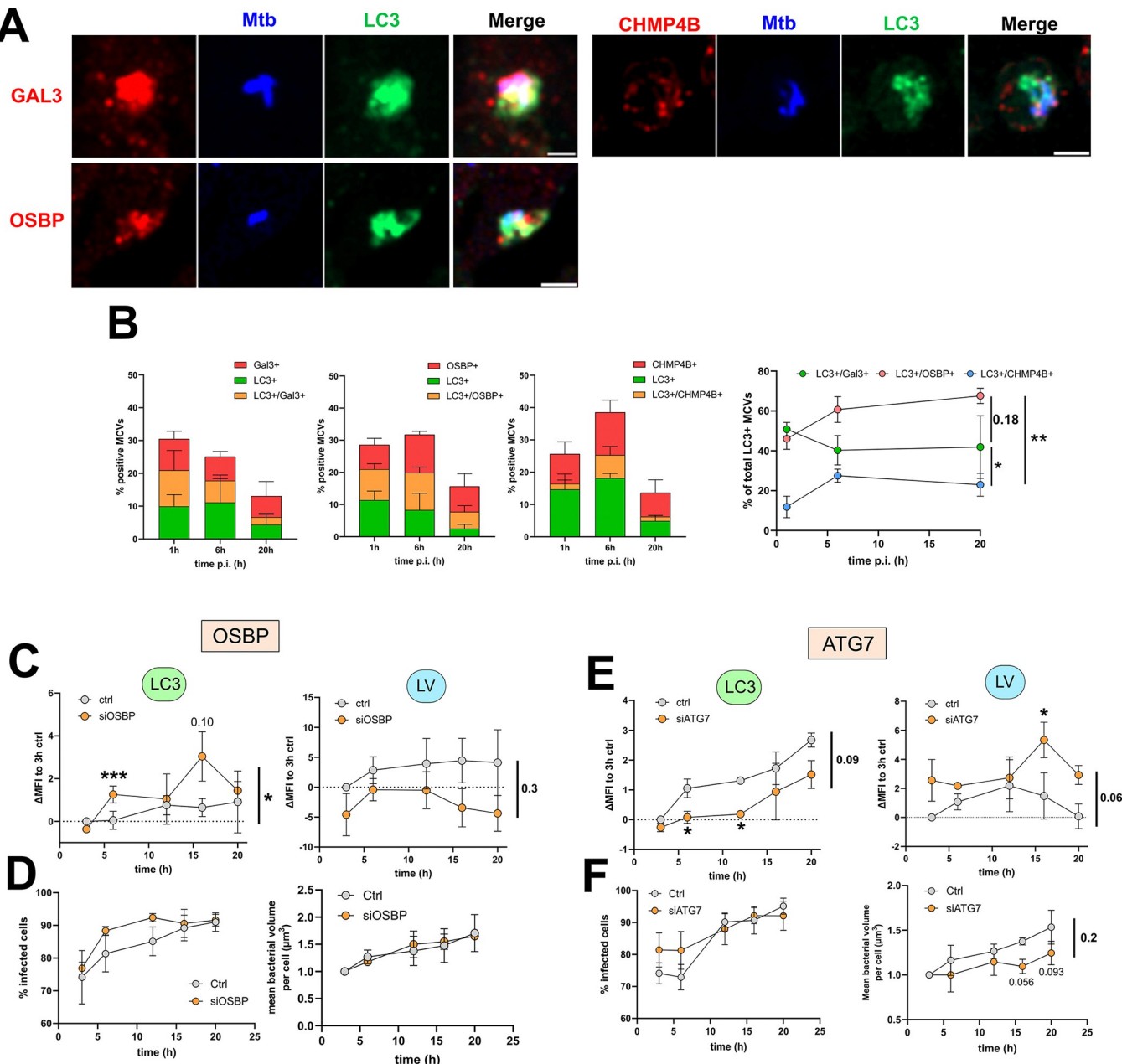

**Fig 6. Autophagy on the MCV correlates with phagosome damage and ER-dependent membrane repair, but does not contribute to membrane repair.**
(A-B) THP-1-GFP-LC3 cells were infected with DsRed expressing H37Rv (Mtb) at an MOI of 10 for 4 h, chased and fixed after an incubation of 1 h, 6 h, and 20 h. The cells were stained either for Galectin-3, CHMP4B or OSBP and imaged by confocal microscopy. (A) Representative images of MCVs being stained by both LC3 and the designated second marker. (B) Quantification of the frequency of association of the markers to the MCV. Points represent the averages of the quantification obtained from 3 independent experiments. The curves were compared using the modified chi-square comparison. $^{*}p \leq 0.05$, $^{**}p < 0.01$. (C-F) THP-1-GFP-LC3 cells were transfected with siRNA against OSBP (C-D) or ATG7 (E-F), infected with Mtb, stained by Lysoview-633 (LV) and imaged by time-lapse confocal microscopy. The movies were analyzed to measure the mean fluorescence intensity (MFI) of GFP-LC3 and LV on single MCV in living cells at the designated time points (C, E). The cells infection rate and the intracellular growth were calculated at the designated time points (D, F). The curves represent the difference of the mean fluorescence intensity (MFI) to their correspondent uninfected 3 h time point, from 3 independent experiments. Each mean was obtained from the measurement on 99–358 MCVs. The difference between time points was individually analyzed by paired t-test, and the curves were compared by modified chi-squared method (see methods).

time in the OSBP-KD cells and the ATG7-KD cells compared to control, or even a decrease in growth in the ATG7-KD (Fig 6D and 6F). This demonstrates that autophagy is not directly involved in the membrane repair observed. The increase in LV MFI (Fig 6E, right) could even indicate that autophagy would compete with the different membrane repair machineries at play.

## Discussion

Our study is the first in depth quantitative analysis of membrane damage and autophagosome formation on single MCVs over time during Mtb infection. For the current work we implemented the novel workflows we recently published [49] for high throughput, single cell analysis of infected cells dynamics, and single MCV quantification of marker recruitment / presence. By applying this workflow to our time-lapse datasets, we were able to track and select different cell populations with distinct observable behaviors, and measure parameters of interest in an automated and limited bias fashion. This was done using open-source and free software that demonstrates the feasibility without having to rely on costly licensed software. As a statement of its flexibility, we also adapted the single cell analysis workflow to automatize the measure of fluorescence at the MCV to provide a limited bias quantification of events than manual scoring (S3 and S6 Figs). This was done in part due to the mobility and amount of MCV per cell preventing the direct use of the single MCV tracking and quantification workflow. This approach allowed to confirm the visual assessment of that the treatment of the cells with IFN-γ had no effect to a decrease in lysoview staining, but no difference in LC3 recruitment. This method was also used on the time-lapse experiments to reveal the dynamics of phagosome maturation and autophagy in the context of ATG7 and OSBP KD. This method was also expanded to classify the MCVs based on their recruitment of markers of interest. It was implemented to decrease the burden of a visual assessment of a large number of MCVs that may increase errors and biases (S6 Fig). Based on our results, the expression of fluorescent LC3 combined with lysoview also provides direct observation of membrane damages induction and membrane repair in a time-lapse microscopy set-up. Thus, these types of markers combined with time-lapse microscopy could be a valuable tool for the future study of membrane repair mechanics and dynamics, during infection of macrophages or other host cells by intracellular pathogens.

Our results combined with published data are summarized in the model shown in Fig 7 which focuses on the MCV and the observed dynamics of LC3 recruitment and Lysoview staining. We observed that MCV acidification frequently precedes LC3 recruitment (Figs 5 and 7). We also observed that LAMP1 was found on most of the MCV as well even without LC3 recruitment, showing that the MCV undergoes maturation (Fig 5F). The MCV's acidification was suggested as a trigger for Mtb to induce membrane damages. Indeed, the Mtb transcriptional regulator PhoP/R system is activated by slightly acidic pH [56] and PhoP/R is a positive regulator of EsxA secretion [57]. In addition, the membrane lytic activity of EsxA is increased by an acidic pH possibly via the pH-mediated dissociation of EsxA from its chaperon EsxB [58–60]. Hence the acidification of the MCV we observe could lead to increased levels of active EsxA within the phagosomal lumen which results in phagosomal membrane damage leading to loss of protons and hence the loss of lysoview staining. However, it was also shown that phagosome maturation blockade is a prerequisite for Mtb phagosome escape [61] and that blocking of phagosome acidification with bafilomycin strongly promotes phagosome membrane damages [61,62]. Accordingly, our data shows that many MCVs did not lose acidification and persisted in this environment for the whole recording (example Movie 13 at https://data.mendeley.com/datasets/cvk8wnfm36/, S7G Fig). Thus, the requirement of acidification to

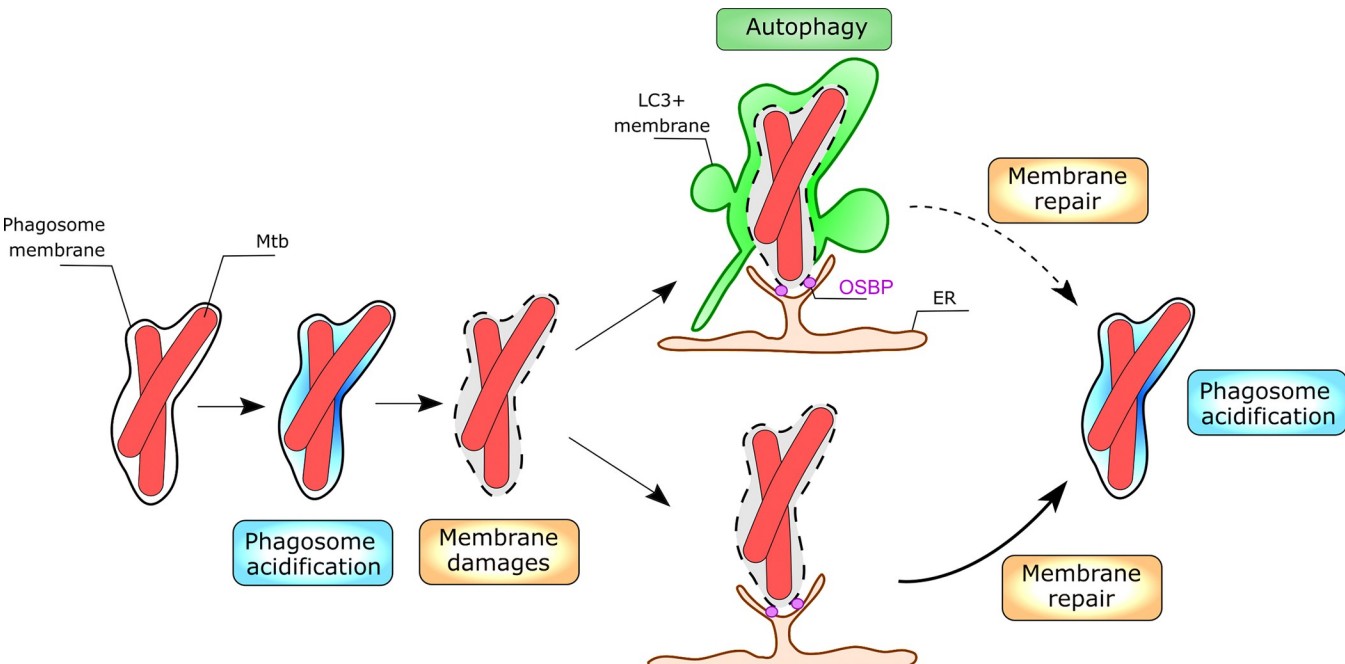

**Fig 7. Model of LC3 recruitment to the Mtb phagosome.** After phagocytosis, the MCV may undergo acidification. This acidification is lost upon induction of membrane damage which triggers LC3 (autophagy) and OSBP recruitment. An ER- but not autophagy-related membrane repair response eventually allows the rescue of acidification of the MCV but can subsequently be damaged again. Alternatively, some damaged might be repaired without concurrent recruitment of LC3. Not shown but some bacteria are eventually able to physically escape from the phagosome or the LC3 positive vacuole to reside in the cytosol.

induce membrane damage to the MCV is still an open question. Additionally, the lysoview probe used in this study can emit fluorescence at pH ≤ 7 according to the manufacturer (Biotium; Cat#70058). It was observed that the phagosome maturation blockade by Mtb stabilize the pH at ~ 6.5 [36,63], which could still potentially be stained by lysoview in our experimental conditions. Thus, we cannot firmly differentiate highly acidic compartments from the phagosome where Mtb at least partially blocked the maturation and consequently additional studies would be required to conclude on the matter.

The membrane damage of the MCV results in a loss of acidity of the MCV as reported by the loss of lysotracker/lysoview staining (Figs 4–7). Lysoview/Lysotracker staining has been used previously as an indicator of membrane damage as it was used to follow lysosome damage and repair mechanisms via live cell imaging [64–67]. Importantly, this method has been used to assess phagosomal membrane damage during Mtb infections as well [34,68]. The time-lapse analysis shows that acidic MCVs lose lysoview staining right before they recruit LC3 (Figs 5 and 7). The rapid and subsequent LC3 recruitment observed here has also been previously associated with membrane damage [17,18,34,46]. To definitively confirm the presence of membrane damages, the cells were stained for GAL3 and the immunofluorescence results demonstrated that LC3 recruitment is accompanied in majority by GAL3 recruitment (Fig 6A and 6B). Previous work on the dynamics of GAL3 and LC3 recruitment suggested that GAL3 recruitment preceded LC3 recruitment [46], and accordingly, GAL3 can stimulate the activation of autophagy [69]. The damages can be explained by the EsxA/ESX-1-induced phagosomal membrane damage described by many other studies, leading to loss of acidity.

Our experimental approach allowed us to determine at what time after phagocytosis MCVs become LC3$^+$ and we noticed that there was a wide range of times post-infection that this event could occur (Fig 2A). Thus, we investigated if LC3 recruitment could be linked to the

number of bacteria per MCV by tracking the bacterial fluorescence signal per MCV and the GFP-LC3 signal over time. Our results showed that there was no correlation between the number of bacteria per MCV and the time of LC3 recruitment (Fig 3A). Quantification of the average number of bacteria per MCV exhibiting early recruitment (< 1 hour) or late recruitment (> 1 hour) also demonstrated no difference (Fig 3B). To the best of our knowledge this is the first time that a detailed analysis and quantification of these parameters was done since the only other time-lapse study following LC3 recruitment onto MCVs aimed to study the structure of LC3$^+$ MCV by CLEM and FIB-SEM imaging [34].

Alternatively to autophagy, the LC3 recruitment on single membranes is proposed to be named atg8ylation and could be mobilized during diverse membrane associated events [70]. The only atg8ylation phenomenon described so far in the context of Mtb infection is LAP [16]. Our study failed to observe signs of LAP being induced after Mtb phagocytosis as the events of LC3 recruitment showed tubulovesicular structure formation, which is induced during canonical autophagy or xenophagy and is distinctively different from LC3 lipidation at the phagosome membrane [71]. We can propose a hypothesis for the lack of detection of LAP in our work. First, Mtb inhibits LAP through the NOX2-inhibiting activity of the Mtb CspA protein [16] and this inhibition might be more potent in human macrophages when compared with BMDMs, in which most of the published work has been performed to date. Another explanation could be the dependency of ROS production for triggering LAP. Indeed, Mtb possesses multiple factors that can neutralize the ROS produced, like KatG [72] and SodA [73], and other factors linked to a decrease in ROS production, such as NuoG [74]. Another study also found that PPE2 can bind p67, a subunit of the NOX-2 complex, to prevent ROS production [75]. In summary, all these factors could prevent LAP and the associated LC3 recruitment. Also, we rarely detected MCVs that were acidified and LC3$^+$. A recent study found that LAP was dependent on V-ATPase recruitment to the phagosome and that the stabilization of the complex by saliphenylhalamide further stimulated the LAP and LC3 recruitment [76]. Hence, we think that the LC3 recruitment to the MCV in our study was almost exclusively associated with an autophagy-related membrane damage repair mechanism and/or autophagosome formation.

The recovery of the acidification after MCV damage then represents membrane repair as the restoration of acidity requires an intact membrane. Our data demonstrates an unexpected reversible property of autophagy and autophagosome formation, as LC3 staining disappeared after the MCV membrane was repaired. This result suggested that autophagy is involved in the membrane repair of the MCV. This hypothesis is in agreement with the role of LC3 in phagosome membrane repair during Salmonella infections [77] and a similar proposed mechanism during *Mycobacterium marinum* infections [25,26]. In addition, deletion of *ATG7* and *ATG14* in human macrophages resulted in a higher fraction of Mtb bacteria in the cytosol, suggesting that autophagy could contribute to membrane repair and keep the bacteria in a phagosome [13]. However, in our experiments, ATG7 KD exhibited opposite effects to OSBP KD, demonstrating that in our cell system autophagy is not directly involved in the repair (Figs 6 and 7). The increase in lysoview staining after knock-down of ATG7 even suggests that autophagy may delay the repair mechanisms potentially by physically competing with access to the MCV (Figs 6 and 7). An explanation could be that autophagy and membrane repair machineries both are activated at the site of membrane damage but that their function is different. Indeed, autophagy is crucial for the removal and recycling of intracellular components, whereas ER-dependent membrane repair machinery restores the compartments integrity [67]. LC3 recruitment to the MCV is dependent upon the Mtb ESX-1 secretion system and requires damage to the phagosome membrane since it was observed galectin-8 recruitment precedes LC3 recruitment to the MCV [6,17,34].

The final aspect of this LC3 recruitment study investigated the potential bactericidal effects of autophagy by examining phagosome/autophagosome acidification levels, both with and without IFN-γ stimulation, which is known to enhance phagosome/autophagosome maturation [78] In contrast to other studies in BMDMs [16], IFN-γ pretreatment did not affect LC3 recruitment and phagosome acidity on average between 1–12 hours post-infection in THP-1 cells. This results shows a potentially crucial difference between mouse and human macrophages and confirms what was reported elsewhere that IFN-γ had no noticeable effect on Mtb fitness in primary human macrophages [5,34] and phagosome maturation [5]. However, in our study, IFN-γ had a strong effect on macrophage viability during infection (S4F Fig). This is most like due to a pathway described in a study showing that stimulating human primary macrophages with IFN-γ promotes a necrotic cell death characterized by DNA release during Mtb infection [79]. Alternatively, our result might be caused by a similar cell death mechanism observed in macrophages stimulated with LPS and IFN-γ, which was able to cause a necrotic type of cell death which is not associated with a protective host response to Mtb infections [80]. Additionally, we observed that while the LC3-TVS did not acidify, both the MCV and the LC3-TVS showed signs of fusion with late endosomes/lysosomes, as characterized by LAMP-1 staining (Fig 5). Thus, it appears that some maturation is occurring but does not translate into observable acidification. This is likely explained by the presence of damages on the LC3-TVS as well, as this structure was also stained with GAL3 (Fig 6A), or previously observed with GAL8 [34]. Acidification of the phagosome or the autophagosome is largely considered as a critical to control or kill intracellular pathogens. Thus, the lack of acidification may show a lack of bactericidal properties that promotes bacterial survival. Monitoring bacterial burden over time in ATG7 knockdown experiments revealed no effect on bacterial burden within the observed time window (Fig 6). A limitation of these experiments is the duration of the recording of 20h which may not be enough time to observe differences in bacterial burden. Accordingly, a previous study showed only a difference in growth of Mtb between WT and *ATG7*-KO in human macrophages at 72 or 96 h.p.i [13]. Additionally, the growth curves provided in this study show a delay of growth by about 24h rather than a killing of the bacteria [13]. It remains unclear whether the observed autophagosomes directly control the infection through an unexplored bactericidal mechanism, indirectly affect it, or lack bactericidal properties altogether. Further studies will be needed to clarify on these points.

## Material and methods

### Reagents and antibodies

Lysotracker blue (Molecular Probes), Lysoview-633 (biotium), Phorbol-Myristate-Acetate (PMA, Sigma), RPMI medium (Gibco), Fetal Bovine Serum (FBS, Gibco), DPBS 1X (Gibco), Recombinant human Interferon Gamma (R&D Systems), OADC supplement (R&D Systems), Zeocin (invivogen), L-Leucyl-L-Leucine methyl ester (LLOMe, Bachem 4000725.0001), anti-ATG7 antibody (Cell signaling technology E6P98), anti-OSBP antibody (Sigma-aldricht HPA039227), anti-Galectin-3 antibody (Cell signaling technology D4I2R XP(R)), anti-CHMP4B antibody (Proteintech, 13683-1-AP), anti-LAMP1 antibody (Cell signaling technology, Rabbit 9091T), Goat anti-mouse AF647 antibody (Jackson laboratory). Goat anti-Rabbit AF647 (Jackson Laboratory), Goat anti-Rabbit HRP (Jackson Laboratory), Goat anti-mouse HRP (Jackson Laboratory). The pEGFP-LC3 (human) plasmid was a gift from Toren Finkel (Addgene plasmid # 24920, http://n2t.net/addgene:24920, RRID:Addgene_24920). The integrative plasmid pYS2:mScarlett for mycobacteria was a gift from Dr. B. Bryson (MIT).

## Mycobacterial strain and culture

The DsRed-expressing strain of *M. tuberculosis* H37Rv was generated by cloning the *dsred* gene from the pMSP12-dsred2 (addgene #30171) into pMAN-1 expression vector previously used [81] by EcoRI and HindIII digestion. The *dsred* gene was amplified using the following primers sequences:

Forward CGGCGAATTCATGGCCTCCTCCGAGAACG;

Reverse GGCTAAGCTTCTACAGGAACAGGTGGTGG

The Mtb strain knocked out for *esxA* (ΔesxA) was described before [82]. The strain was made fluorescence by electroporating it with the plasmid pYS2:mScarlett.

Bacteria were grown in liquid Middlebrook 7H9 medium supplemented with 10% oleic acid-albumin-dextrose catalase (OADC) growth supplement, 0.2% glycerol, 0.05% tween 80, and Zeocin 100μg/mL. For infections, the bacteria were washed in 0.05% PBS-tween 80 before being added to the infection medium (see procedure below).

## Cell line culture, differentiation and infection

THP-1 cell lines expressing GFP-N-terminal LC3B fusion protein used in a previous study [50], were provided by the lab of Dr. John Kehrl (NIH) and prepared using a plasmid (pMXs-IP-GFP-LC3) provided by the lab of Dr. Noboru Mizushima (University of Tokyo). The cells were cultured in RPMI 1640 (ATCC modification) supplemented with 10% heat-inactivated Fetal bovine serum (complete medium). The cells were differentiated with 20 ng/ml of PMA for 20–24 hours. In the case of IFN-γ pre-stimulation, the cells were washed in 1X DPBS and incubated overnight with 10 ng/mL of IFN-γ in complete medium. Stimulation efficiency was evaluated by flow cytometry and by the staining of MHC-II (HLA/DR) surface expression (S3 Fig).

Peripheral blood mononuclear cells (PBMCs) from healthy blood donors were isolated from leukopaks by using Ficoll density gradient centrifugation. For monocyte isolation, 10x106 PBMCs per well were seeded in a 6 wells plate for 2 h in unsupplemented RPMI. The attached monocytes were washed three times with unsupplemented RPMI and incubated for 7 days in RPMI 1640 supplemented with human off-the-lot serum 5% and M-CSF at 10 ng/ml (hMDM complete medium). On day 7, the cells were detached using trypsin for transfection (see below).

For infections of THP-1 cells, the cells were washed in 1X DPBS and incubated in RPMI supplement with 5% normal human AB serum (infection medium). For hMDMs, after a similar wash step, the cells were incubated in hMDM complete medium. The bacteria were added to the cell at an MOI of 2 for THP-1 cells, or MOI 1 for hMDMs without a washing step for time-lapse imaging or an MOI 10 for 4 hours (THP-1 cells) for immunofluorescence and confocal imaging. After the 4-hour incubation, cells were washed and incubated for 16 hours in complete medium supplemented with 100 μg/mL Gentamycin.

## hMDM transfection

The cells were transfected using the Amaxa 4D nucleofector X unit (Lonza) following the protocol provided by the manufacturer (https://bioscience.lonza.com/download/product/asset/21595). Briefly, the macrophages in suspension were pelleted and resuspended in Nucleofector solution from the P3 Primary cells 4D nucleofector X kit at a density 8x10$^5$ cells per transfection. 3μg of GFP-LC3 plasmid were added and the cells were electroporated in a 100μL nucleo-cuvette using the DP-148 program. Then the cells were diluted into fresh medium and aliquoted in uncoated polymer-bottom observation chambers (ibidi). The cells were incubated for 24 hours before infection and time-lapse confocal imaging.

## Knock-down of ATG7 and OSBP

The procedure for the nucleofection of siRNA was done following the neon transfection manufacturer's instructions:

(https://assets.thermofisher.com/TFS-Assets/LSG/manuals/neon_device_man.pdf). Briefly, the THP-1 cells were differentiated as described above and then detached using the Cell stripper (Corning) solution. The cells were then pelleted and resuspended in 100uL of the E2 buffer that is part of the Neon transfection reagents kit (thermofisher). 100pmoles of siRNA for ATG7 (s20650, Thermofisher), OSBP(11742, Thermofisher), or negative control siRNA (#1, 4390843, Thermofisher) were added and the cells were electroporated using two 20ms pulses at 1400V. The cells were directly incubated in complete culture medium for 48 hours in glass-bottom observation chambers (ibidi) at $2x10^5$ cells/well.

## Western blot

After the 48h of incubation, an amount of $5x10^5$ transfected cells were lysed with lysis buffer (1% NP-40, 0.4 mM EDTA, 10 mM Tris-Hcl, 150 mM NaCl) for 2 minutes at room temperature. The proteins were dosed using the BCA protein dosage kit (thermofisher) and according to the manufacturer protocol. Then 10 µg of lysate was mixed with Laemmli buffer 1X (BioRad) with β-mercaptoethanol 10% and boiled at 95˚C for 5–7 minutes. The proteins were separated by SDS-PAGE in a 12% polyacrylamide gel (Genscript) at 200V for 30 min. The proteins were then transferred on a nitrocellulose membrane. The membrane was then washed with DPBS + tween20 0.1% (PBST) and block with PBST-milk 5% for 30 min at room temperature. The membrane was then incubated with PBST-milk 0.5% + anti-OSBP (1:1000) or anti-ATG7 (1:1000) antibody + anti-β-actin (1:5000) at 4˚C overnight. The membrane was washed with PSBT three times and then incubated with PBST-milk 0.5% + goat anti-mouse HRP 1:15000 + goat anti-rabbit HRP 1:15000 for 2h at room temperature. The membrane was washed three times with DPBS and revealed using with SuperSignal West Pico ECL kit (34577, Thermo Scientific).

## Time-lapse confocal microscopy

For the THP-1 cells, they were seeded and differentiated in glass-bottom observation chambers at $2x10^5$ cells/well. The cells were stimulated with IFN-γ, if needed, before the infection (see above). For imaging of phagosome maturation, the infection medium was supplemented with Lysoview-633 at a final concentration of 1:2000 in infection medium and left on the cells during recording. The cells were imaged using a Zeiss laser scanning confocal microscope LSM-800, equipped with two gallium arsenide phosphide photomultiplier tube (GaAsP-PMT) detectors and a transmitted light photomultiplier tube detector (T-PMT), using the 63x/NA1.4 Oil objective. The observation chamber was maintained at 37˚C and supplied with 5% CO2 through a humidifier system. A 10 µm z-stack (1 µm steps) was acquired every 10 minutes on 4 to 8 fields of view over 16 to 20 hours. Each field of view was 1024 x 1024 or 2048 x 2048 pixels corresponding to 101.81 x 101.81µm and 202,83 x 202.83 µm, respectively.

To assess phagosome acidity using lysotracker blue, the cells were infected in glass bottom observation chambers at an MOI of 10, as described above, and incubated for 16 hours. The cells were washed three times with 1X DPBS and incubated in complete medium supplemented with 1:10000 lysotracker blue for 30 minutes. The cells were washed, incubated in complete culture medium, and imaged by confocal microscopy.

## Immunofluorescence and confocal microscopy

The THP-1-GFP-LC3 cells were seeded in a 24 wells plate with glass coverslips at the bottom, at a density of $5x10^5$ cells / well. The cells were differentiated as described and infected for 4h at an MOI of 10. The cells were washed three times with DPBS and incubated in complete medium supplemented with Gentamycin at 100 µg/mL for 20h. For the test of lysosome damages, the cells were treated with LLOMe at 1µM for 45 min. Then, the cells were washed three times with DPBS and fixed in PFA 4% for an hour at room temperature. After fixation, the cells were washed three times with DPBS and incubated twice for 5 minutes with DPBS supplemented with Glycine (100mM). The cells were washed once with DPBS and then blocked with DPBS + BSA 0.2% + Saponin 0.05% (PBSAP) + FcBlock TruStain 1:100 (BioLegends) for 30 minutes. The cells were incubated anti-Galectin-3 (1:200), anti-OSBP (1:80), anti-LAMP1 (1:200), or anti-CHMP4B (1:100) in PBSAP at 4˚C overnight. They were then washed three times with PBSAP and then incubated with PBSAP + goat anti-mouse or goat anti-rabbit AF647 antibodies (1:200) for 2h at room temperature. The cells are finally washed with PBSAP three times and then mounted on slides using Prolong gold anti-fade with DAPI. The cells were imaged on a Zeiss LSM 980 Airyscan 2 confocal microscope equipped with a 63X oil-immersion objective.

## Image analysis

All the MCV qualitative assessments of LC3 recruitment, fluorescence quantifications, single cell tracking and fluorescence quantification, and single MCV fluorescence quantification were performed using the ImageJ/Fiji software [83] directly using custom macros or in python using the library PyimageJ [84].

For the quantification of frequency of recruitment in Fig 1, and the assessment and categorization of LC3 recruitment in Fig 2A, the MCV were visually scored as positive or negative. The method for manual tracking and quantifying the fluorescence at the MCV over time in Figs 2A and 5 was previously described [85]. For the single cell tracking and quantification of bacterial burden over time, this was done using the da_tracker workflow [49]. For the quantification of LC3 volume over time, the bacterial burden section of the workflow was adapted to measure volume of signal in the LC3 channel instead of the bacterial channel.

For single MCV fluorescence quantification at defined time points, the workflow is illustrated in S1 Fig and ran in PyimageJ adapted from the da_tracker workflow. The source code is provided in supplementary material. In more details, the cells were segmented using Cellpose on the T-PMT channel. The bacteria channel is then isolated and the extracellular signal as well as signal in dead cells were removed. The bacterial centroid is then detected and collected using the plugin Trackmate [86]. The bacteria were also detected using Fiji particle detection function and the generated ROIs were used to measure the centroid coordinates. Then, the coordinates of the centroids collected from Trackmate were compared to the coordinates of the ROIs centroid collected from Fiji using the nearest neighbor function from geopandas library in python. The refined ROI set representing was then used to measure the mean fluorescence intensity of the channels of interest. The code for this single MCV workflow is provided as supplementary material. For the determination of the relative bacterial burden at these defined time points, the bacterial fluorescence was analyzed to calculate the bacterial volume per cell, using the BBQ method described elsewhere [87]. For quantifying the recruitment of markers on MCVs shown in Figs 5F and 6A–6B, the single MCV quantification approach was adapted and extended as illustrated in S6 Fig. Starting with the detection of MCV centroids using TrackMate and Fiji, resulting ROIs were identified and expanded incrementally by 1-pixel, resulting in concentric ring areas extending outward from the MCV

boundary. Each ring has a width of 1 pixel, allowing for fluorescence intensity measurements at increasing distances from the MCV. By measuring the fluorescence intensity on the MCV and around it at each ring, a curve representing fluorescence distribution as a function of distance from the MCV in pixels is generated. This approach enables a spatially resolved quantitative analysis of fluorescence around individual MCVs. The process is repeated for each detected MCV and for each marker analyzed. The fluorescence curves are normalized by subtracting their respective mean values to adjust for baseline intensity differences, ensuring that variations reflect spatial distribution rather than absolute intensity. Next, the normalized curves are grouped into 10 clusters using agglomerative hierarchical clustering in Python (scikit-learn library). The choice of 10 clusters optimally differentiates distinct curve shapes and prevents clusters from containing mixed patterns. Finally, the clusters are classified as either 'positive' or 'negative' for the analyzed marker, based on the curve shape: curves that display a decreasing signal intensity from the MCV to the periphery, reaching a plateau, are designated as positive, while those without this pattern are categorized as negative. The code provided also allows for visual curation of clusters and, in cases where mixed patterns are observed, for subclustering. If a cluster contains curves with heterogeneous shapes, the user can visually inspect these mixed-pattern clusters and, if necessary, reapply the clustering process specifically to refine these groups. This step ensures that each cluster represents a distinct and consistent fluorescence pattern. For the visual curation, the code generates a montage of images for each channel, displaying each detected MCV in a single, scrollable window to facilitate easy review of clustering results.

## Data representation and statistics

Data were plotted using Graphpad Prism 9 or 10 except for the density map in Fig 5E, which was created using R, and in S7 Fig using python. Perplexity AI was puntually used to improve clarity of the text. The figures were designed using Powerpoint and Inkscape, and the model design was made using Inkscape. All statistical tests were performed using Prism or excel. The correlation between two variables was tested using Pearson's correlation test. Statistical differences between two groups were tested using paired Student $t$-tests or more than two groups using a one-way ANOVA followed by Dunnett *post-hoc* statistical test. The difference between two entire curves from time-lapses was done using the modified Chi-squared method on an Excel spreadsheet provided here [88]. Comparisons with a *p*-value greater than 0.05 were considered significant.

## Supporting information

**S1 Fig. Quantification of autophagy induction over time.** Single cell tracking and quantification of LC3 volume over time plotted on a 3 variables graph with the bacterial volume as a color dimension. Each dot represents the measurement in 1 cell in 1 frame. (A) Result of the quantification in cells exhibiting at least one temporary LC3 recruitment on at least one MCV ($N_{cells} = 41$, $N_{points} = 4299$). (B) Quantification in cells that don't exhibit LC3 recruitment ($N_{cells} = 11$, $N_{points} = 1222$). (C) Quantification in uninfected cells ($N_{cells} = 32$, $N_{points} = 3349$). (TIF)

**S2 Fig. Comparison between Lysoview-633 and Lysotracker blue staining on infected cells.** (A) Representative images of MCVs in THP-1- GFP-LC3 cells negative (-), positive for Lysoview 633 (Lyso+), for LC3 (LC3+), and double positive (LC3+Lyso+). (B) quantification of the fraction of MCVs in THP-1 cells stained for lysoview 633 or lysotracker blue. (C) The cell death level was quantified by the adenylate Kinase (AK) release assay on THP-1- GFP-LC3

treated or not with Lysoview 633 1:2000 for 48h. RLU = relative luminescence unit.
(TIF)

**S3 Fig. Diagram of the image analysis workflow implemented for the fluorescence quantification on single MCV.** The cells are first segmented using Cellpose. The ROIs obtained are applied on the bacterial channel to retain the signal only in living cells. The bacterial signal is then segmented on Fiji to create ROIs and measure the centroids coordinates, and analyzed using Trackmate to obtain the coordinates of the MCVs' centroids. Their distance to the ROIs centroids is calculated to retain the closest using the nearest neighbor calculation. The selected ROIs were finally applied on the channels of interest and the mean fluorescence intensity is finally measured.
(TIF)

**S4 Fig. Effect of IFN-γ stimulation on MHC-II expression and cell death.** THP-1 cells were treated with IFN-γ 5 ng/mL, 10 ng/mL overnight or left untreated. Cells were fixed and stained for MHC-II expression using HLA/DR antibody (BD). Some Treated cells with 10 ng/mL were stained with isotype control. The staining was analyzed by flow cytometry. (B) Representative time lapse showing the untreated cells (top panel) or pre-stimulated with IFN-γ (bottom panel). Time stamp format is hh:mm. Chevrons are showing cells presenting the ballooning phenotype. (C) The cell death frequency was quantified by counting the proportion of cells exhibiting the ballooning phenotype during infection. Data are from 4 independent experiments. Data points color indicate the same experiment. Data between untreated and IFN-γ treated groups were compared by paired $t$-test. $**p < 0.01$.
(TIF)

**S5 Fig. Fluorescence traces of LC3 and lysoview of individual MCVs.** Compilation of the LC3 and lysoview fluorescence quantification on individual MCVs isolated from 4 independent experiments. Only the MCV exhibiting LC3 recruitment for more than 3 h were retained.
(TIF)

**S6 Fig. Analysis workflow for the classification of MCVs.** From multi-channel image, the MCVs are detected using Trackmate and Fiji as described in S1 Fig. The obtained ROI around the MCV (noted 1) is then enlarged by 1 pixel. The new ROI is enlarged by 1 pixel and the operation is repeated until 5 concentric ROIs are created around the original MCV ROI. The total fluorescence can be measured in between concentric circles to create a curve of fluorescence representing the distribution of fluorescence on and around the LCV. Each curve is then normalized by subtracting their respective average value. The pool of curves is then classified into 10 clusters using agglomerative hierarchical clustering. The cluster are then defined as positive or negative based on the shape of the cluster. Curves showing a decreasing fluorescence intensity to a plateau value were considered positive for the marker. Curves showing a flat or increasing fluorescence intensity were considered negatives for the marker. The analysis was repeated for each marker analyzed on the image. The positivity and negativity for each marker per MCV was then determined.
(TIF)

**S7 Fig. The LC3 recruitment is EsxA-dependent and the interplay between LC3 recruitment and loss of acidification is observed in primary human macrophages.** (A-C) hMDMs expressing GFP-LC3 were stained with Lysoview-633, infected with DsRed expressing Mtb at an MOI 1 and imaged by time-lapse confocal imaging for 20 h. (A) Representative image of an infected macrophage at different time point. The tracked MCV are being magnified and annotated. (B) Quantification of fluorescence at the MCV for each time point. Mean fluorescence intensity (MFI) of GFP and Lysoview (LV) was measured and normalized between values of 0

and 100 as 0 the minimum value recorded, and 100 the maximum value. (C, left panel) Quantification of LV and LC3 normalized MFI and was compared. Each dot represents the GFP and LV normalized MFI at one time point for one MCV from 2 independent experiments (n = 10 MCVs). (C, right panel) The distribution of data points in left panel is shown as a density map. (D-G) THP-1-GFP-LC3 cells were infected with Mtb WT or ΔesxA bacteria at an MOI 2 and imaged by time-lapse confocal imaging for 20 h. (D) Representative images of infected cells by Mtb WT or ΔesxA at 6 h post infection (E) Single cell tracking of infected cells and quantification of LC3 volume over time. The points represent the average LC3 volume per cells, and the ribbon the standard error to the mean (N = 2, $n_{(cells)}$ = 47 for WT and $n_{(cells)}$ = 32 for ΔesxA. (F) Representative frames of MCVs from WT (top) and ΔesxA (bottom) showing LC3 recruitment. The associated normalized MFI of GFP-LC3 and LV was plotted (right). (G) MCVs from WT and ΔesxA that were trackable for at least 3 h that did not show LC3 recruitment were followed and the LC3 MFI was quantified on the MCV. The graph displays the means and standard error to the mean (N = 2, $n_{(WT)}$ = 10, $n_{(ΔesxA)}$ = 12). The difference between the curves was analyzed by modified chi-squared method. ***$p<0.01$, ****$p<0.001$.
(TIF)

**S8 Fig. Validation of GAL3, OSBP and CHMP4B antibodies by inducing lysosome damages using LLOMe.** Differentiated THP-1-GFP-LC3 cells were treated with LLOMe at 1μM for 45 minutes, fixed and stained with antibodies against the indicated target. The cells were imaged using confocal microscopy. The figure displays representative images of cells untreated or treated with LLOMe.
(TIF)

**S9 Fig. Relative abundance of ATG7 or OSBP in the THP-1-GFP-LC3 cells after knockdown with siRNA.** The cells were lysed, and the lysate was analyzed by SDS-PAGE followed by western blot. The β-actin blotting served as a loading control. (top) Western blotting results from cells that were left untreated, treated with negative control siRNA, and siRNA against ATG7. The lysates correspond to the cells that were used for infection in Fig 6C–6D. (Bottom) Western blotting results from cells untreated, treated with negative control siRNA, and siRNA against OSBP. The lysates correspond to the cells that were used for infection in Fig 6E–6F. A low exposure and a high exposure are shown for clarity in the relative abundance of OSBP compared to the β-actin loading control.
(TIF)

**S10 Fig. Evolution of relative bacterial volume per cell over time in OSBP and ATG7 knock-down cells.** THP-1-GFP-LC3 cells were transfected with siRNA against OSBP (left) or ATG7 (right), infected with Mtb, stained by Lysoview-633 and imaged by time-lapse confocal microscopy. The relative bacterial burden was determined by calculation of the bacterial volume per cell (see methods). These results correspond to the raw values presented in the Fig 6D and 6F. The dots correspond to the average bacterial volume per cell and the color corresponds to the different independent experiments.
(TIF)

**S1 Code. Code ran to analyze the fluorescence intensity on single MCV at single time points, used for Figs 4D–4E, 6C, and 6E.** The workflow is also illustrated in S3 Fig.
(PDF)

**S2 Code. Code ran to analyze the recruitment of markers around MCVs from immunofluorescence images in Fig 5F and 6B.** The workflow is also illustrated in S6 Fig.
(PDF)

The movies referenced in this article were deposited here: https://data.mendeley.com/datasets/cvk8wnfm36/. Or can be obtained using the download links below: Movie 1, Movie 2, Movie 3, Movie 4, Movie 5, Movie 6, Movie 7, Movie 8, Movie 9, Movie 10, Movie 11, Movie 12, Movie 13.

## Acknowledgments

We thank Dr Adrien Schahl for his insights in the optimization of the cell tracking calculation, and Dr Guillaume Ferré for his feedback on data analysis. We also thank Dr Ashton T. Belew for his feedback on the optimization of the python code in the different workflows.

## Author Contributions

**Conceptualization:** Jacques Augenstreich, Volker Briken.

**Data curation:** Jacques Augenstreich.

**Formal analysis:** Jacques Augenstreich, Anna T. Phan, Anushka Poddar.

**Funding acquisition:** Volker Briken.

**Investigation:** Jacques Augenstreich, Anna T. Phan, Hanzhang Chen.

**Methodology:** Jacques Augenstreich, Charles N. S. Allen, Anushka Poddar, Volker Briken.

**Project administration:** Jacques Augenstreich, Volker Briken.

**Resources:** Jacques Augenstreich, Charles N. S. Allen, Anushka Poddar, Lalitha Srinivasan.

**Software:** Jacques Augenstreich, Anushka Poddar.

**Supervision:** Jacques Augenstreich, Volker Briken.

**Validation:** Jacques Augenstreich, Volker Briken.

**Visualization:** Jacques Augenstreich, Anna T. Phan.

**Writing – original draft:** Jacques Augenstreich.

**Writing – review & editing:** Jacques Augenstreich, Charles N. S. Allen, Volker Briken.

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
