## [Decision Letter · Decision Letter 0]

27 May 2024

Dear Dr. Briken,

Thank you very much for submitting your manuscript "Dynamic Interplay of Autophagy and Membrane Repair During Mycobacterium tuberculosis Infection" for consideration at PLOS Pathogens. As with all papers reviewed by the journal, your manuscript was reviewed by members of the editorial board and by several independent reviewers. In light of the reviews (below this email), we would like to invite the resubmission of a significantly-revised version that takes into account the reviewers' comments. The revised submission should provide new data from experiments that the reviewers have suggested in order to strengthen the manuscript and fully support the conclusions. In addition, to increase clarity, a substantial rewriting of the manuscript is recommended. 

We cannot make any decision about publication until we have seen the revised manuscript and your response to the reviewers' comments. Your revised manuscript is also likely to be sent to reviewers for further evaluation.

Sincerely,

Padmini Salgame

Academic Editor

PLOS Pathogens

Michael Otto

Section Editor

PLOS Pathogens

Michael Malim

Editor-in-Chief

PLOS Pathogens

orcid.org/0000-0002-7699-2064

Reviewer's Responses to Questions

**Part I - Summary**

Reviewer #1: In this manuscript entitled Dynamic Interplay of Autophagy and Membrane Repair During Mycobacterium tuberculosis Infection by Augenstreich and colleagues, the authors investigate the dynamics of LC3 recruitment to mycobacterial-containing endolysosomes and how the this complex subcellular interaction impacts (auto)phagolysosomes biogenesis, membrane damage and repair, and ultimately Mtb-induced cell-death.

By using a stable LC3-GFP-expressing THP-1 cell line, and a Mtb-DsRed fluorescent reporter strain, the authors monitor by live-cell imaging in BSL-3 condition, the spatiotemporal dynamics of LC3 recruitment onto the Mtb-containing vacuole over a 16h-20h period. By analysing 73 independent events, they show that LC3 recruitment is highly heterogeneous after phagocytosis and that the observed events can be subclassified in 3 categories based on their distinct colocalization profiles (e.g. early recruitment, late recruitment or multiple rounds of recruitments).

Quantitative analysis suggests that there is no correlation between LC3 recruitment and the time after infection, or the intracellular bacterial burden of each MCV. Additional experiments confirmed such results but also suggested that if higher intracellular bacterial burden did not directly correlate with LC3 colocalization, it was associated with increased recruitment rates. Investigations of the biological role/function of such recruitment on Mtb-containing endolysosomes using image-based tools showed that LC3 recruitment is not dependent of IFN-Ɣ pre-treatment and does not drive acidification. On the contrary, results presented by the authors suggest that LC3 positivity and endolysosomal acidification were mutually exclusive. Finally, the authors investigate whether LC3 recruitment contributes to membrane damage and/or repair by combining 4-colours immunofluorescence imaging and knocking-down the expression of the key genes OSBP and ATG7. Based on their findings, they proposed a biological model in which autophagy does not play a major role in cell autonomous defence against Mtb nor the process of membrane repair in human macrophages.

Deciphering the cellular and molecular bases responsible for Mtb intracellular growth restriction/permissiveness is crucial. Moreover, the development of innovative, high-resolution quantitative imaging, especially relying on live-cell technologies in BSL3 environments is very challenging and demanding but highly relevant to study dynamic processes such as the induction and dynamics of xenophagy. In that context, the work performed by the authors in this study is of primary interest for the host-pathogen interaction community and the Mtb field, and therefore the broad readership of PLOS Pathogens.

Overall, this paper is interesting and well-written, but unfortunately it is difficult to clearly identify a significantly novel and original contribution to the field arising from this study, since it lacks a functional/mechanistical understanding of the observations reported by the authors. In addition, several conclusions are not appropriately supported by the data presented in this manuscript. Several experiments would require more thorough validation steps and complementary investigations/analyses to fully support the claims of the authors.

Therefore, I would recommend the authors to clarify some information and consider the following points/suggestions to strengthen their manuscript and further consolidate their data in order to fully support their conclusions, which might increase its impact for our community.

Reviewer #2: In Augenstriech et al. the authors monitor the dynamics of LC3 recruitment to Mtb-containing phagosomes in human THP1 cells using live cell imaging. They find that LC3 recruitment is frequent and often transient, and moreover, phagosome acidification and LC3 recruitment often occur independently and sequentially, rather than in tandem as would have been expected. These findings will be of interest to the TB field and are important in terms of better understanding the dynamics of autophagy-related pathways during Mtb infection, although the manuscript could have gone further in trying to obtain mechanistic insight into what is driving these interesting dynamics. Overall, the experimental design, imaging, and analysis are well done, and my main concern is related to occasional over interpretation/overstatement of the data, as described below.

Reviewer #3: In the manuscript “Dynamic Interplay of Autophagy and Membrane Repair During Mycobacterium tuberculosis Infection” by Augenstreich et al, the authors use live cell imaging to study the recruitment of autophagy markers and the acidification of phagosomes during Mtb infection. They find that LC3 recruitment a dynamic process that occurs heterogeneously and transiently during Mtb infection, and LC3 recruitment is mutually exclusive with phagosome acidification. In addition, they show that OSBP-dependent membrane damage repair participates in the repair of LC3+ MCVs, allowing some to re-acidify and shed their LC3.

This study uses live cell imaging to examine the dynamics of autophagy and membrane repair during Mtb infection. This is a critical question the field as a majority of work thus far has used fixed cells to examine discrete time points. By using live cell imaging, this study provides critical insight into the dynamic recruitment and frequent loss of markers during infection, which informs our models of how host cells detect and respond to intracellular Mtb. It also demonstrates/provides key recourses/workflows that can be used by other groups to perform similar analyses examining other host factors/pathways of interest. The authors’ also Discussion does an especially good job contextualizing their findings with what’s been previously published and acknowledging some limitations of their approaches.

While the authors provide convincing data that autophagy and membrane damage are dynamic processes in Mtb-infected THP-1 cells, it’s not clear how universally applicable this finding is to other macrophage cell lines or to primary macrophages. To firmly draw the conclusions in this manuscript, key observations should also be made in other macrophage types. In addition, some of their conclusion regarding the antibacterial activity of autophagy don’t seem to be fully supported by their data.

**Part II – Major Issues: Key Experiments Required for Acceptance**

Reviewer #1: Major Point(s)

First, I would like to thank the authors for making accessible their “single_bact_quant” source code. However, it is not very clear in their manuscript how LC3 positivity was determined and which threshold value was used (Fig.1-Fig.2). Indeed, in Fig.1 & Fig.2, the authors report heterogeneity in LC3 recruitment over time, but information is lacking about how a LC3 positive MCV was determined. What was the fluorescence cut off to determine LC3(+) or LC3(-) MCV? How was it calculated? When looking at Panels 2B-C & 2D-E, the Mean Fluorescence Intensity (MFI) associated to the MCV can range from 15 to 100 MFI and still considered by the authors as positive. Can the authors comment about this? Moreover, adding an extra (dashed/coloured) line on the Y-axis of their graphs (Fig2) to highlight where the positive threshold has been set would be great to facilitate the readers understanding.

In Fig.3, the authors claim that there is no correlation between LC3 recruitment, the time post-phagocytosis and the bacterial burden. There are several issues with such claim.

First, the R2 evaluates the scatter of the data points around the fitted regression line, but does not indicate anything about statistical significance (Fig3A). An appropriate p-value should be calculated to determine whether the “no correlation” observed is statistically significant or just random. Could the authors please look into this.

Second, from the Material and Methods section it seems that the current resolution, is not high enough to allow the authors to identify/segment single-bacterial cell and therefore claiming that the authors can calculate the number of bacteria per MCV seems to be overstated.

Indeed, the authors mention the following statement “The cells were imaged using a Zeiss laser scanning confocal microscope LSM-800, equipped with two gallium arsenide phosphide photomultiplier tube (GaAsP-PMT) detectors and a transmitted light photomultiplier tube detector (T-PMT), using the 63x/NA1.4 Oil objective. The observation chamber was maintained at 37oC and supplied with 5% CO2 through a humidifier system. A 10 μm z-stack (1 μm steps) was acquired every 10 minutes on 4 to 8 fields of view over 16 to 20 hours.”

Can the author comment about the X, Y and Z resolution achieved with their microscopy system, eventually by looking at the properties/metadata of their acquired images? Can they really achieve single-bacterial cell segmentation within infected macrophages specially when looking at Z-stack distant from each other of 1µm, and not reconstituting Z-projection? Can they comment about how was this done?

To the reviewer’s knowledge, achieving automated single-bacterial cell resolution on intracellular mycobacteria has not been reported in the literature so far, and therefore would be a real Tour de Force from the authors. Can you comment on this and provide all the details require to support such conclusion? This is important for the entire community, and people developing innovative quantitative imaging approaches. The use of bacterial area (if analysed in 2D) or volume (if analysed in 3D using maximum projection) expressed in pixel2/µm2 or pixel3/µm3 on the other end seems more appropriate.

In their Fig4A, the authors first investigate the potential relationship between LC3 recruitment and subcellular acidification, and report that LC3 positivity and Lysotracker Blue positivity are mutually exclusive.

Here is their statement: “Strikingly, at 24 hours post-infection, none of the LC3+ MCVs were positive for LTB, although about 15% of MCVs were positive for LTB (Figure 4B). These results strongly suggested that LC3 recruitment does not result in Mtb contained within autophagolysosome”.

However, based on the data presented in Fig4-AB, it seems that the dual-positive event was observed and quantified. In addition, the authors report 15% of LysoTracker positivity in the text, but between 25-30% seem more appropriate based on the graph 4B. As previously, request for LC3, can the author comment on LysoTracker positivity how positivity was defined?

To support their claim, regarding the lack of maturation in such compartment, have the authors considered performing some additional control experiments with alternative functional proteolytic probes such as DQ-BSA or BMV109 or even staining for V-ATPase or Cathepsins?

In Fig4C, it is not clear what the authors wanted to report. “THP-1-LC3-GFP cells were treated with IFN-γ 10ng/mL or left untreated overnight and infected with DsRed expressing H37Rv at MOI 2. Cells were also stained with Lysoview-633 (LV) and the dye was kept during image acquisition. Cells were image by time lapse confocal microscopy”. Why the authors do not report each individual channels? It is impossible to see LC3 or LV positive event with such representation. Please could the authors consider an alternative presentation of their images? It is also not clear why the authors only reported data at 6h post infection, where they have performed live-imaging over 20 hours. Could they add more timepoints, or even considered replotting their data highlighting the dynamics in the system rather than just focusing on one timepoint? How the 6 hours was chosen? Some representation as displayed in Fig5, for both IFN-gamma treated or the control condition, would be more appropriate to illustrate their findings and conclusions.

Finally, the title of Fig. 4 is “Autophagosomes are not bactericidal and IFN-γ treatment has no effect on LC3 recruitment frequency or phagosome maturation”.

There is to the reviewer knowledge no evidence that autophagosomes are bactericidal, and since only LysoTracker blue staining was performed, it seems a bit overstated to conclude that there is no maturation.

As suggested previously, an additional dashed or coloured Y-line could be added in Fig5B-D and Y and X in FigE to indicate positivity. Please indicate here as well, how positivity was defined. If some rare events would appear with both LC3 and LysoView, as suggested by the positions of some individual dots in Fig 5E left panel, it would be interesting to select some of these peculiar events, and display some of their features by live-cell imaging in a main or supplementary figure, with access to the movies. These rare events, might by underrepresented but could be of broad interest for the community and therefore I would encourage the authors to display more about it.

In Fig6, the authors investigate whether LC3 positive structures might be as well positive for galectin3, ALIX or OSBP, three markers widely used for more damage and repair processes. It is not clear why the authors did not perform this assay overtime, knowing that LC3 positivity fluctuates overtime, and starts very early during the infection kinetics. Can the authors comments about this?

Interestingly, at this specific time point, a larger number of LC3 (+) structures were reported to be associated with Galectin3 or OSBP (~2-5 fold), whereas very few LC3-ALIX double positive events were observed. Have the authors considered looking at the ESCRT-III effector CHMP4 as well or other repair markers? Knowing that the temporal recruitment/dynamics of these markers can be interconnected and eventually precede one to each other, I would recommend the author to look into earlier additional timepoints (2hours pi, 6hours pi and 24hours pi) and eventually try to perform some additional live-cell imaging in transfected THP1-LC3 GFP cells to further support their conclusion. In the recent years, the concept of minor lysosomal lesion vs rupture has appeared. In that context, several protein actors are involved, and here a more thorough investigation of which one is involved, and when; seems to be crucial to really delve into LC3 positivity and its function in concert with other actors during Mtb infection/pathogenesis.

Finally, if the WB presented in FigS6 are convincing, a functional validation of the knock down would have been important to really validate that the pathway is not functional. This could be autophagic flux experiments, or immunofluorescence upon specific stimulation to further confirm that the Knockdown are effective before conducting experiments on Mtb. This validation step is lacking, and its difficult to trust the findings/conclusion of the Mtb experiments without the appropriate validation. In addition, pushing the replication towards 48h-72h would be interesting as monitoring growth/infectivity and cell-death over 24hours, is very limiting. The authors could really benefit from performing longer experiments and exploring additional timepoint to observe some events later in the kinetics.

Overall, it is an interesting story that requires some additional control experiments, some clarifications regarding the methodologies and approaches used, and I would personally recommend the authors to focus more onto the mechanistical aspects (from the host or the pathogen) of such recruitment and repair rather than just some observations/descriptions, as this would greatly benefit the findings for our community.

Reviewer #2: 1. The authors should consider that change in phagosome acidification might occur for reasons other than phagosomal damage (e.g. H+ can be consumed in luminal reactions, altered by counter-ion flux, leakage, neutralized by the bacilli, etc.…see PMID: 33505976). If they use a bacterial mutant that is defective in Esx-1 function, what are the dynamics of acidification?

Reviewer #3: The authors draw conclusions based on experiments performed in only THP-1s, a macrophage-like immortalized cell line that doesn’t always faithfully replicate macrophage responses to Mtb. To firmly establish the validity of these results, the authors should perform key live cell imaging experiments in additional macrophage-like cell lines or primary macrophages. While use of human macrophages is obviously preferable, much work on macrophage responses to Mtb has been performed using mouse macrophages, so utilizing mouse macrophages in order to assess primary cells, use fluorescent reporters, or establish that findings are generalizable would be reasonable.

I’m not convinced the authors establish through their data that “there is no major role of autophagy in cell autonomous defense against Mtb” since they do not directly examine Mtb viability or examine time points beyond 20 hr of imaging when Mtb killing might occur. I think this conclusion of the paper needs to be softened to better align with the data presented, or additional data supporting this claim should be included.

The authors perform a careful analysis of the timing of initial recruitment events, but based on the Methods, it seems infections are not synced via spinfection. As a result, it’s hard to establish the time of phagocytosis if bacteria could be phagocytosed at various times post-infection as they settle by gravity onto the macrophages. The authors should be mindful of this caveat when describing the timing of events in the Results.

Figure 1 was cut off in the submission PDF, so the data associated with it could not be evaluated.

**Part III – Minor Issues: Editorial and Data Presentation Modifications**

Reviewer #1: Minor Points

- Please add the appropriate reference for the Gobal TB report from the WHO.

- Please include the exact procedure that was carried out for nucleofection within the Material and Methods section instead of redirecting towards “The procedure for the nucleofection of siRNA was done following the neon transfection manufacturer’s instructions”. Also include the reference of the siRNA that have been used for OSBP and ATG7 since their respective references are missing.

- In panel Fig4. D-E-F it is not clear what the authors did not report the standard deviation as before?

- Same comment regarding the Fig S7?

- The use of one unique font used for the main figures throughout the manuscript is recommended as for some figures multiples fonts are mixed.

Reviewer #2: 1. I don’t think the authors can make the conclusion, as stated, in their abstract, “Our data suggest that there is no major role of autophagy in cell autonomous defense against Mtb nor membrane repair.” They looked at bacterial burden only up to 20 hours, and the pH change is an indirect measure that they are interpreting as reflecting repair.

2. The conclusion that ALIX is not recruited to Mtb phagosomes requires that the ALIX antibody works in their conditions—i.e. it would be important to see a positive control that ALIX is recruited in their cell type, with their fixation method, etc. after lysosomal damage. Also, the findings on ALIX and OSBP are just from a late time point, so may not speak to the earlier events (e.g. 1-6 hours in Fig. 2). Finally, their discussion blurs the distinction between minor and more substantial damage which might require different repair mechanisms.

3. Since LC3-positivity always follows acidification, consider the role of the V-ATPase in non-canonical autophagy processes in which raised pH drives V-ATPase assembly and ATG16L1 recruitment (PMID: 36288298). More mechanistic insight could be gained around these observations, for example, by blocking acidification.

4. How do the authors envision that LC3 is rapidly lost from phagosomes, and how would that differ in egress vs exclusion. When the authors say, “quickly diffused away,” could they be more specific? How long, on average, does that take? Is it different for egress vs exclusion? Do the dynamics provide any insight into how this might happen? Consider PMID: 33686057

5. There are some inconsistencies with the literature – e.g. on the role of Atg7 in repair, that aren’t explained or fully investigated (e.g. corroboration with additional methods of blocking autophagy, etc.). The impact of IFN-� is another example, which might be related to cell type and the substantial cell death that

6. . In the images (Fig. 6A), OSBP signal appears to be within the LC3+ signal. Is this expected?

7. The discussion is a bit long and difficult to read.

Reviewer #3: Several sections of the Results paper were difficult to read and understand due to typos or overly clunky language. The authors should consider editing the text with an eye toward clarity so that the reader can more easily understand the experiments, the rationale, the analysis, etc.

The results from IFN-g treatment are surprising, and in examining the microscopy images, it seems the IFN-g-treated cells have far more bacteria in the fields shown. Is the increased cell death simply due to differences in MOI?

It’s not clear why ATG7 was chosen for KD to test if autophagy is required for membrane repair. Are other ATGs also detrimental to membrane repair? Is this detrimental effect of autophagy specific to Mtb infection (and bacterial virulence strategies) or does it also occur with sterile membrane damage?

The authors refer to their cells as expressing LC3-GFP, which implies the GFP is fused to the C-terminus of LC3. If this is the case, it interferes with LC3 function. As such, most reporters are GFP-LC3. The authors should clarify in the text and the Materials & Methods.

In Fig 4B and similar figures, the colors are too similar to one another, and the ordering of colors in the bar graph is not consistent with their or with their order in the legend, making it difficult to read and interpret.

The authors use the phrase “minimal to none” when the data indicates that “minimal” is the accurate description; they should not say “none” if in fact there is some.

The abstract conveys mixed messages about the involvement of and requirement for autophagy in membrane repair; I think this could be edited to be clearer and more consistent with the data presented.

It may be beyond the scope of this manuscript, but it might be worth mentioning in the Discussion that LC3 is the terminal marker of autophagy; it would be interesting to examine the dynamics of intermediate markers of autophagy, like p62/NPD52, or sensors of membrane damage like galectin-3.

PLOS authors have the option to publish the peer review history of their article (what does this mean?). If published, this will include your full peer review and any attached files.

Reviewer #1: No

Reviewer #2: No

Reviewer #3: No
---

## [Decision Letter · Decision Letter 1]

11 Dec 2024

Dear Dr. Briken,

We are pleased to inform you that your manuscript 'Dynamic Interplay of Autophagy and Membrane Repair During Mycobacterium tuberculosis Infection' has been provisionally accepted for publication in PLOS Pathogens.

Before your manuscript can be formally accepted you will need to complete some formatting changes, which you will receive in a follow up email. A member of our team will be in touch with a set of requests. Also, please address the minor modifications suggested by Reviewer 1. 

Best regards,

Padmini Salgame

Academic Editor

PLOS Pathogens

Michael Otto

Section Editor

PLOS Pathogens

Sumita Bhaduri-McIntosh

Editor-in-Chief

PLOS Pathogens

orcid.org/0000-0003-2946-9497

Michael Malim

Editor-in-Chief

PLOS Pathogens

orcid.org/0000-0002-7699-2064

Reviewer Comments (if any, and for reference):

Reviewer's Responses to Questions

**Part I - Summary**

Reviewer #1: The revised manuscript entitled “Dynamic Interplay of Autophagy and Membrane Repair During Mycobacterium tuberculosis Infection” by Augenstreich et al. has undergone significant revisions and improvements since the initial review. I would like to thank the authors for addressing most of comments and suggestions, resulting in a solid piece of work. The amount of data is important and the conclusions drew by the authors are supported by most of the presented data. The revised manuscript stands out as an excellent contribution to the field of host-pathogen interactions in tuberculosis.

Strengths: The authors have effectively addressed the feedback provided in the earlier review. The revisions have significantly strengthened the overall quality of the manuscript. The language and presentation have been refined, making the manuscript clearer and more accessible to a broader audience. The inclusion of additional data and explanations has improved the comprehensiveness of the work.

Reviewer #2: The authors' findings will be of interest to the TB field and are important in terms of better understanding the dynamics of autophagy-related pathways during Mtb infection. They adequately addressed my questions, and I have no further concerns.

Reviewer #3: The revised manuscript by Augenstreich et al provides a great deal of important live cell imaging data, revealing novel aspects of macrophages’ cell-autonomous responses to Mtb infection. The additional data provided by the authors and the edits to the text have improved the manuscript and yielded an exciting, impactful, and important study. The manuscript provides many intriguing observations that will fuel future mechanistic studies in the field.

**Part II – Major Issues: Key Experiments Required for Acceptance**

Reviewer #1: No major revisions are required since the authors have adequately addressed most of my previous comments.

Reviewer #2: (No Response)

Reviewer #3: None

**Part III – Minor Issues: Editorial and Data Presentation Modifications**

Reviewer #1: While the manuscript is already of high quality, the following minor points could be considered for further improvement:

- I have previously highlighted that the authors used multiple types of fonts in their Figures, which could be improved by using only Arial or Helvetica.

The authors claimed that they addressed this comment however, we can see that fonts are still mixed in their figures. Therefore, I recommend the authors to carefully address this and avoid mixing Calibri with Arial etc.

- In Fig3B, the authors do not clearly identify the number of bacteria. Therefore, I would recommend them to label the Y-axis with “Estimated number of bacteria/MCV” as claiming that they know exactly the number of bacteria is wrong.

- Fig5E should be name as “Dot-Plot” and not “Dot-Blot”

Reviewer #2: Line 383-- typo

Reviewer #3: None

PLOS authors have the option to publish the peer review history of their article (what does this mean?). If published, this will include your full peer review and any attached files.

Reviewer #1: No

Reviewer #2: No

Reviewer #3: No

---

## [Editor Report · Acceptance letter]

24 Dec 2024

Dear Dr. Briken,

We are delighted to inform you that your manuscript, "Dynamic Interplay of Autophagy and Membrane Repair During Mycobacterium tuberculosis Infection," has been formally accepted for publication in PLOS Pathogens.

Best regards,

Sumita Bhaduri-McIntosh

Editor-in-Chief

PLOS Pathogens

orcid.org/0000-0003-2946-9497

Michael Malim

Editor-in-Chief

PLOS Pathogens

orcid.org/0000-0002-7699-2064